# Error Forcing in Recurrent Neural Networks

**A. Erdem Sağtekin**
New York University
University of Tübingen
aes10217@nyu.edu

**Colin Bredenberg**
University of Montreal
Mila - Quebec AI Institute
colin.bredenberg@mila.quebec

**Cristina Savin**
New York University
csavin@nyu.edu

## Abstract

How should feedback influence recurrent neural network (RNN) learning? One way to address the known limitations of backpropagation through time is to directly adjust neural activities during the learning process. However, it remains unclear how to effectively use feedback to shape RNN dynamics. Here, we introduce error forcing (EF), where the network activity is guided orthogonally toward the zero-error manifold during learning. This method contrasts with alternatives like teaching forcing, which impose stronger constraints on neural activity and thus induce larger feedback influence on circuit dynamics. Furthermore, EF can be understood from a Bayesian perspective as a form of approximate dynamic inference. Empirically, EF consistently outperforms other learning algorithms across several tasks and its benefits persist when additional biological constraints are taken into account. Overall, EF is a powerful temporal credit assignment mechanism and a promising candidate model for learning in biological systems.

## 1   Introduction

Most theoretical neuroscience models posit that learning in the brain occurs via feedback projections that guide local synaptic plasticity [1–12]. Such models—based on approximations of gradient descent—assume that feedback signals are delivered without interfering with network activity, for example by multiplexing feedforward and feedback signals [13], via global neuromodulatory signals [14], or by only infinitesimally perturbing the network state [15, 16]. However, circuit-level evidence of a crisp separation between learning and dynamics is less clear. Moreover, there is substantial evidence that humans are able to learn both cognitive and motor skills with very few feedback examples [17, 18], and that they can rapidly correct for systematic errors induced by various manipulations (e.g. external force fields [19] or visuomotor manipulations [20]). This suggests that the brain may use dynamic error-correcting mechanisms to affect within-trial neural activity in order to rapidly improve performance before longer-timescale synaptic learning and consolidation [21, 22]. However, previous models of dynamic error correction-based synaptic plasticity in the brain have either been used in exclusively feedforward neural network architectures [23, 24], or have been used in recurrent neural networks provided with dense feedback signals [25–27], without rigorous quantification of the temporal credit assignment capabilities of the proposed learning algorithms.

Here, we introduce *error forcing* (EF), an algorithm designed to exploit both greedy error-based guidance of circuit dynamics *and* temporal credit assignment to stabilize learning of long-term temporal dependencies (Fig. 1a). Improving on generalized teacher forcing [28], we show that our algorithm can better stabilize learning for supervised tasks with sparse feedback and long temporal horizons. Further, we justify our approach through a theoretically principled connection to Bayesian

39th Conference on Neural Information Processing Systems (NeurIPS 2025).

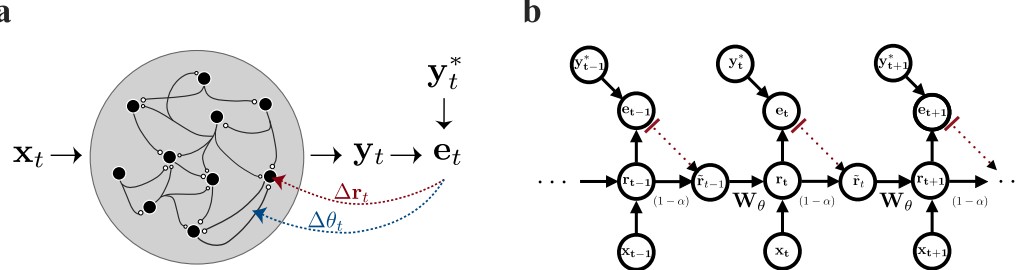

Figure 1: **a)** Illustration of the error forcing mechanism. Errors are used to update the synaptic weights given by any gradient-based algorithm (blue dashed-lines), as well to update the activity (red dashed lines). **b)** Discrete-time computational graph of the EF, showing only the forward changes. Red dashed lines depict the error to activity projection weights. Notice we used stop-gradients (small red line), which prevents gradients flowing backwards from activity to errors. See Section 2.3 for mathematical justification of this step.

inference and variational expectation maximization [29], demonstrating that dynamic error correction has a sensible connection to existing optimization algorithms. Lastly, we demonstrate that EF provides significant benefits for temporal credit assignment across a variety of tasks, and that its biologically plausible extensions preserve similar properties. These results collectively show that EF has promise as both an improvement for temporal credit assignment in machine learning methods, and as a model of rapid adaptation in the brain.

## 2 Learning by error forcing

### 2.1 Background

Consider the general formulation of a discrete-time RNN with a linear readout:

$$\mathbf{r}_t = \mathbf{F}_\theta(\mathbf{r}_{t-1}, \mathbf{x}_t), \tag{1}$$

$$\mathbf{y}_t = \mathbf{W}_\phi \mathbf{r}_t, \tag{2}$$

where $\mathbf{r}_t \in \mathbb{R}^N$ is the network state, $\mathbf{x}_t \in \mathbb{R}^{N_x}$ the input, and $\mathbf{y}_t \in \mathbb{R}^{N_y}$ the output. Parameters $\theta$ and $\phi$ define the recurrent dynamics and readout, respectively. Minimizing the error between network outputs and the target $\mathbf{y}_t^*$ requires computing the derivative of the loss with respect to network parameters $\theta$, typically achieved by backpropagation through time (BPTT). Following [30], given a loss function $\mathcal{L} = \sum_{t=T_s}^{T} \mathcal{L}_t(\mathbf{y}_t, \mathbf{y}_t^*)$, where $T_s < t < T$ is defined as response period, the BPTT equations are given by:

$$\frac{\partial \mathcal{L}}{\partial \theta_i} = \sum_{t=1}^{T} \frac{\partial \mathcal{L}_t}{\partial \theta_i} \quad \text{with} \quad \frac{\partial \mathcal{L}_t}{\partial \theta_i} = \sum_{t'=1}^{t} \frac{\partial \mathcal{L}_t}{\partial \mathbf{r}_t} \frac{\partial \mathbf{r}_t}{\partial \mathbf{r}_{t'}} \frac{\partial \mathbf{r}_{t'}}{\partial \theta_i}, \tag{3}$$

$$\frac{\partial \mathbf{r}_t}{\partial \mathbf{r}_{t'}} = \frac{\partial \mathbf{r}_t}{\partial \mathbf{r}_{t-1}} \frac{\partial \mathbf{r}_{t-1}}{\partial \mathbf{r}_{t-2}} \cdots \frac{\partial \mathbf{r}_{t'+1}}{\partial \mathbf{r}_{t'}} = \prod_{t \geq i > t'} \frac{\partial \mathbf{r}_i}{\partial \mathbf{r}_{i-1}} = \prod_{t \geq i > t'} \mathbf{J}_i \tag{4}$$

Depending on the spectral properties of the Jacobians, the above product can lead to instabilities, where error signals either decay or grow without bounds [30]. For example, this occurs when the network state changes little over time (i.e. $\mathbf{J}_i$ are very small across steps, as when the network is visiting a stable attractor). Such conditions are of practical importance in many neuroscience tasks with an explicit memory component [31], and here the Jacobian product can exponentially decay, leading to a vanishing gradient problem. Conversely, when the goal of the RNN training is to generate chaotic dynamics, the gradients inevitably explode [32].

This well-known issue of vanishing and exploding gradients motivated the development of alternative approaches such as teacher forcing (and its variations) [28, 33–37], especially for the self-supervised

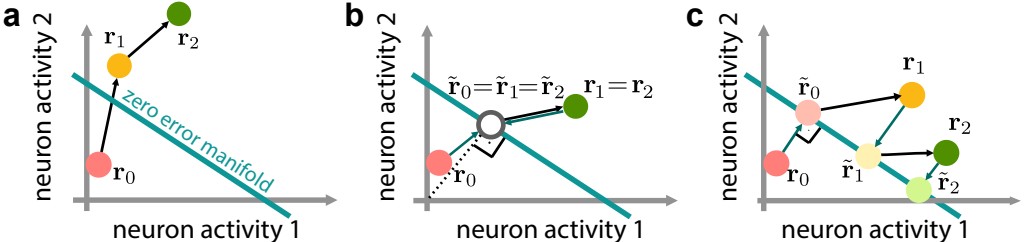

Figure 2: Geometric explanation of differences between no forcing (a), teacher forcing with the minimum-norm solution (b), and error forcing (c); $\alpha = 1$. **a)** Each circle denotes the RNN state $\mathbf{r}_t$ at a given time, and arrows denote the application of the forward function (Eq. 1). The RNN freely runs during training when no forcing mechanism is used. The zero-error manifold is the set of points where, for a given set of readout weights (which do not change within a trial), the network produces the target outputs $\mathbf{y}_{\mathbf{t}}^*$ (assumed constant for visual simplicity). **b)** The teacher forcing mechanism results in choosing the target state as the minimum-norm solution (white circle, the closest point to the origin on the zero error manifold). Beginning from $\mathbf{r}_0$, we apply forcing to get $\tilde{\mathbf{r}}_0$ via Eq. 5. Then, we apply the forward function via Eq. 6 which results in $\mathbf{r}_1$. The same procedure will be applied over the trial, and since the teacher state is not a function of the RNN state, it will remain the same, causing the RNN to repeat the same activity sequence. Since $\tilde{\mathbf{r}}_0 = \tilde{\mathbf{r}}_1 = \tilde{\mathbf{r}}_2$, we illustrate these points with one circle. **c)** When error forcing is used instead, the RNN can explore the phase-space during training.

setting. In its generalized version [38, 28], teacher forcing (TF) pushes neuron activity towards states that would correspond to correct outputs:

$$\tilde{\mathbf{r}}_t := (1 - \alpha)\mathbf{r}_t + \alpha\mathbf{r}_t^*, \tag{5}$$

$$\mathbf{r}_t = F_\theta(\tilde{\mathbf{r}}_{t-1}, \mathbf{x}_t), \tag{6}$$

where $\mathbf{r}_t$ denotes the natural RNN state, $\mathbf{r}_t^*$ is the target (teacher) that satisfies $\mathbf{y}_{\mathbf{t}}^* = \mathbf{W}_\phi\mathbf{r}_{\mathbf{t}}^*$, and their linear interpolation leads to the forced dynamics $\tilde{\mathbf{r}}_t$, with $0 \leq \alpha \leq 1$. When $\alpha = 0$, the method reduces to BPTT. When states are partially forced, the computational graph of the network dynamics changes, which in turn changes the BPTT Jacobians:

$$\mathbf{J}_t = \frac{\partial \mathbf{r}_t}{\partial \mathbf{r}_{t-1}} = \frac{\partial \mathbf{r}_t}{\partial \tilde{\mathbf{r}}_{t-1}} \frac{\partial \tilde{\mathbf{r}}_{t-1}}{\partial \mathbf{r}_{t-1}} = \frac{\partial F_\theta(\tilde{\mathbf{r}}_{t-1})}{\partial \tilde{\mathbf{r}}_{t-1}} \frac{\partial \tilde{\mathbf{r}}_{t-1}}{\partial \mathbf{r}_{t-1}} = (1 - \alpha)\tilde{\mathbf{J}}_t. \tag{7}$$

Well-behaved gradients can thus be ensured by a judicious selection of $\alpha$ [28].

## 2.2 Geometric perspective

The question at hand becomes: How should we determine the target states $\mathbf{r}_t^*$? When the readout is invertible, we can simply use its inverse to go from the output space to the neural space: $\mathbf{r}_t^* = \mathbf{W}_\phi^{-1}\mathbf{y}_t^*$. Similar approaches have been shown to be beneficial for RNN training when the output is higher dimensional than the hidden neurons, or when the effective latent dimensionality is lower than the output dimensionality [28, 33]. However, in most practical scenarios, and, arguably, in the brain [39], the output is low-dimensional but the readout pools over a larger group of neurons in the circuit. Mathematically, it follows that there is an entire manifold of neural states with zero error. In this case, it is not *a priori* clear towards which target state we should force neural activity.

In one version of TF, the activity target is defined as the minimum-norm mapping from the low-dimensional output space to activity space, given by the readout pseudoinverse, $\mathbf{r}_t^{*\text{min}} = \mathbf{W}_\phi^+\mathbf{y}_t^*$. To illustrate the potential issue with this strategy and a better way forward, we will start by a simple toy example. Consider the scenario of a network with 2 neurons and a 1-dimensional output. Considering the separation of time-scales between neuron activities and synaptic updates, we can think of the mapping between activity and output as constant within trials. The only factor that changes the zero-error manifold within trials is the output target (hence the offset of the manifold). For visualization purposes only, let us assume constant target states $\mathbf{y}^*$ (Fig. 2, cyan line), so that the zero error

manifold does not move from time step $t$ to $t + 1$ within a trial. When states are not forced, the RNN runs freely, finding itself outside of the zero-error manifold (Fig. 2a). With full forcing ($\alpha = 1$), using the minimum-norm solution, the RNN is driven to the teacher state, and the next step follows Eq. 6. For a stationary target output, the same teacher state is used at each step, causing the RNN to repeat the same activity pattern (Fig. 2b), limiting the exploration of the neural space during learning. Even when the output is time-varying, the teacher states will always lie on the one-dimensional subspace orthogonal to the null space of the readout (therefore orthogonal to the zero error manifold). Thus, traditional TF enforces unnecessarily strong constraints on the network's activity, which, as we will show empirically, can slow down the learning process and damage test performance.

In contrast, our proposed EF selects the optimal state such that $\mathbf{r}_t^*$ is the orthogonal projection of $\mathbf{r}_t$ onto the manifold of possible optimal responses (Fig. 2c). The idea is that instead of enforcing a minimal norm solution, we select the minimal *intervention* to the preexisting network state to minimize the error, which should result in a broader exploration of the zero-error manifold over learning and generally less forcing of the dynamics away from their natural state.

More precisely, the EF target state is given by $\mathbf{r}_t^* = \mathbf{r}_t + g(\mathbf{W}_\phi^+ \mathbf{e}_t)$, where $g(\cdot)$ is a 'stop gradient' operation. We will justify the use of the stop gradient here in terms of our Bayesian perspective on EF (Section 2.3; Appendix B), but we note that the empirical performance of methods with and without the stop gradient was similar (Appendix C). The resulting computational graph is shown in Fig. 1b. Substituting this term into Eq. 5,

$$\tilde{\mathbf{r}}_t = (1 - \alpha)\mathbf{r}_t + \alpha \left( \mathbf{r}_t + g(\mathbf{W}_\phi^+ \mathbf{e}_t) \right), \tag{8}$$

$$= \mathbf{r}_t + \alpha g(\mathbf{W}_\phi^+ \mathbf{e}_t), \tag{9}$$

where $\mathbf{e}_t = \mathbf{y}_t^* - \mathbf{y}_t$. Given this form of the full dynamics (including the feedback loop), we can use BPTT (or any of its bio-plausible approximations, as explained in Section 2.4) for the synaptic weight updates. When the EF mechanism is used with BPTT, we called the resulting learning procedure EF-BPTT.

To see how EF changes the BPTT gradient computation and how it is different from the changes induced by teacher forcing, we can write Eq. 9 by rearranging the terms as (see Appendix A):

$$\tilde{\mathbf{r}}_t = \mathbf{I}\mathbf{r}_t - \alpha g(\mathbf{W}_\phi^+ \mathbf{W}_\phi \mathbf{r}_t) + \alpha \mathbf{W}_\phi^+ \mathbf{y}_t^*. \tag{10}$$

This form illustrates the differences between how EF and TF modify dynamics and gradient updates. EF and TF become *dynamically* equivalent when $N_y \geq N$ as the $\mathbf{W}_\phi^+ \mathbf{W}_\phi$ term becomes an identity matrix (see Appendix A), but the stop gradient operation prevents the TF gradient dampening phenomenon (Eq. 7) because the two rightmost terms of Eq. 10 have zero gradient. When $N_y < N$, EF only immediately affects activity in the row space of the readout. Since it doesn't restrict activity in the readout nullspace (see Fig. 2c), neuron activities are free to move along the zero error manifold. These observations imply that the empirical performance improvements produced by EF relative to TF and BPTT are not due to dampening of Jacobian magnitudes, but are rather due to minimally guiding network activity towards an optimal activity regime.

## 2.3 Bayesian perspective

Although we have provided a geometric interpretation of why EF may be expected to improve performance relative to TF, a question remains: is there a way to connect EF to theoretically principled optimization methods? The need for such a link is underscored by a well-known limitation of teacher forcing: it is unclear why TF should work at test time, since the training objective (an RNN with forced states) differs from the one we ultimately care about (unforced states). In fact, there has been targeted work on this train–test mismatch (e.g., [40, 41]). For GTF in particular, annealing the parameter $\alpha$ during training has been as a way to get around this discrepancy. In contrast, we will see that error forcing does not introduce this train–test mismatch, because it can be linked to a principled optimization method in which training-time forcing can be expected to generalize to unforced dynamics at test time.

One way construct this connection is by viewing EF as a mechanism to train an input-conditioned latent generative model of the output targets, where the error corrections $\boldsymbol{\epsilon}_t$ correspond to hidden latent variables. In this view, latent error corrections are *inferred* during training, conditioned on

both input stimuli $\mathbf{x}$ and output targets $\mathbf{y}$; subsequently, in an approximation to the variational Expectation-Maximization algorithm [29], network parameters are updated via gradient descent. Under this interpretation, test-time performance can be viewed as a stimulus-conditioned generative process, where both latent errors and output targets are sampled via a stochastic process.

To illustrate how this works, we first replace the deterministic RNN dynamics (Eq. 1) with a Gaussian state-space model (by the addition of Gaussian noise $\boldsymbol{\epsilon}_t$ onto the hidden state and $\boldsymbol{\eta}_t$ onto the output):

$$\mathbf{r}_t\big(\boldsymbol{\epsilon}_{0:t}, \mathbf{x}_{0:t}, \theta\big) = F_\theta\Big(\mathbf{r}_{t-1}\big(\boldsymbol{\epsilon}_{0:t-1}, \mathbf{x}_{0:t-1}, \theta\big), \mathbf{x}_t\Big) + \boldsymbol{\epsilon}_t \tag{11}$$

$$\mathbf{y}_t = \mathbf{W}_\phi \mathbf{r}_t\big(\boldsymbol{\epsilon}_{0:t}, \mathbf{x}_{0:t}, \theta\big) + \boldsymbol{\eta}_t, \tag{12}$$

where, as in the deterministic case $\mathbf{r}_t$ refers to the network hidden state, $\mathbf{y}_t$ refers to the output targets, and here we are treating $\boldsymbol{\epsilon}_t$ as an additive latent variable. There are two factors that differentiate this model from the more traditional Kalman Filter: 1) the transition function $F_\theta(\cdot)$ is nonlinear; and 2) the error $\boldsymbol{\epsilon}_t$ is treated as the random variable for inference, whereas the hidden state variable $\mathbf{r}_t$ is treated as a deterministic function that depends on all network parameters $\theta$, as well as preceding errors $\boldsymbol{\epsilon}_{0:t}$ and stimuli $\mathbf{x}_{0:t-1}$. We show (Appendix B) that greedily inferring the optimal state space correction $\boldsymbol{\epsilon}_t^*$ given current and preceding targets $\mathbf{y}_{0:t}$, stimuli $\mathbf{x}_{0:t}$, and corrections $\boldsymbol{\epsilon}_{0:t-1}^*$ via *maximum a posteriori* filtering produces nearly identical state space dynamics as the deterministic EF case, while still allowing for BPTT learning (via the reparameterization trick [42]). In this case, "reparameterization" refers to viewing the errors $\boldsymbol{\epsilon}_t$ as stochastic latent variables, which allows for the use of backpropagation through time through the deterministic hidden state update $F_\theta(\cdot)$. The stochastic EF dynamics are given by:

$$\tilde{\mathbf{r}}_t = \bar{\mathbf{r}}_t + \mathbf{W}_\phi^\top \left( \mathbf{W}_\phi \mathbf{W}_\phi^\top + \frac{\sigma_\eta^2}{\sigma_\epsilon^2} \mathbf{I} \right)^{-1} \mathbf{e}_t, \tag{13}$$

where $\bar{\mathbf{r}}_t$ is the "uncorrected" state driven by network inputs and recurrent corrections, $\mathbf{e}_t$ is the error in output space, and $\tilde{\mathbf{r}}_t$ is the "corrected" state. The state space error correction $\boldsymbol{\epsilon}_t^* = \mathbf{W}_\phi^\top \left( \mathbf{W}_\phi \mathbf{W}_\phi^\top + \frac{\sigma_\eta^2}{\sigma_\epsilon^2} \mathbf{I} \right)^{-1} \mathbf{e}_t$ corresponds to the output space error $\mathbf{e}_t$ multiplied by a regularized pseudoinverse of the decoder, much as we have used in our original deterministic model. The main difference is that instead of controlling the magnitude of forcing via $\alpha$, the ratio between latent noise and observation noise variances $\sigma_\eta^2/\sigma_\epsilon^2$ controls the magnitude of forcing [33] ($\sigma_\eta^2/\sigma_\epsilon^2 \to 0$ gives full forcing and $\sigma_\eta^2/\sigma_\epsilon^2 \to \infty$ gives no forcing).

This perspective connects EF to inference in state-space models (e.g., the Kalman filter) and provides a theoretically grounded basis for stochastic EF training. In particular, it explains why backpropagating through the error signal that modulates the RNN dynamics is unnecessary—hence the use of the stop-gradient operator—because the variational EM algorithm performs coordinate descent: it first infers the errors given the current parameters and then updates the network parameters while treating the inferred errors as constants (see Appendix B for details). Furthermore, it clarifies why applying error forcing only during training is natural (as opposed to also forcing at test time): during training, error corrections correspond to target-conditioned latent states inferred via approximate MAP estimation under the generative model, whereas at test time both the latent errors (zero-mean, low-variance Gaussian noise) and the targets are generated by the model (Eq. 11).

## 2.4 Bio-plausible error forcing

In this section we describe additional approximations that we explored to increase the biological plausibility of EF, which in its most basic form is unrealistic due to its dependence on BPTT [43] and violations of weight transport [44] in its feedback signals used for both learning and error correction.

To alleviate issues associated with BPTT, we use RFLO [45] for temporal credit assignment in place of BPTT. RFLO provides a local, online eligibility trace, by ignoring non-local interactions when computing gradients. We allow weight transport when computing synaptic updates for two reasons: the original work reported similar performance with and without random feedback weights, and recent studies show that local learning rules can adapt feedback pathways to approximate the transpose or pseudoinverse of the feedforward weights [46, 47]. We refer to this variant—using our EF with RFLO-based temporal credit assignment—as EF-RFLO.

Less explored in previous work is the role played by feedback state-forcing weights. In particular, EF introduces an additional weight transport problem because the optimal forcing weights are constrained to be the pseudoinverse of the feedforward decoder weights, and it is unclear how this type of parameter information could be shared between physiologically separate synapses. To investigate this issue, we explored random fixed forcing weights and random learned forcing weights, pairing these feedback mechanisms with RFLO synaptic learning. These experiments probe how essential the exact pseudoinverse feedback matrix is to produce the benefits afforded by EF.

## 3 Numerical results

In this section, we evaluate the effectiveness of the error forcing (EF) mechanism and compare it with competing algorithms. We consider a diverse battery of tasks: some are ethologically relevant and widely used in computational neuroscience (evidence integration), and some are difficult long-time credit assignment benchmarks (nonlinear working memory). Each task family includes hyperparameters controlling the task difficulty to further probe the computational efficiency of EF. We first present results for EF-BPTT, followed by EF-RFLO and its variants.

### 3.1 EF-BPTT

First, we tested the algorithms on the delayed XOR task (Fig. 3a). This is a nonlinear working memory task in which two input channels deliver rectangular pulses of amplitude $+1$ or $-1$, separated by a fixed inter-stimulus delay at the start of each trial (Fig. 3a). The network must compute the XOR of the two stimuli and report the answer after a second delay. Crucially, this second delay varies across trials; consequently, the response time also varies across trials. To mark the start of the response window, we provide an explicit cue signal—if the second delay were fixed, such a cue would be unnecessary. Task difficulty is controlled by the *mean* of the second delay (not by its trial-to-trial variability): for each task condition, the second delay is sampled from a distribution with a specified mean and limited jitter, and increasing this mean increases the memory demand and thus the task difficulty. The network loss is assessed only during the response time, and so the RNN readout is unconstrained during non-response periods. Therefore, the forcing mechanism is also only active during the response period.

For every algorithm, and for every task difficulty, we trained 20 networks (continuous time RNNs, details in Appendix D) and monitored their convergence. Specifically, a network is said to have converged if its mean-squared error (MSE) remains under 0.1 for at least 10 gradient steps, within a maximum of 200 training epochs. While the exact setting of this convergence threshold is somewhat arbitrary, it does not qualitatively affect the nature of our results. The optimal degree of forcing $\alpha$ was determined by a grid search (note that $\alpha = 0$ reduces EF-BPTT to vanilla BPTT).

While task performance decreased sharply for standard BPTT with increasing task difficulty, EF-BPTT maintained a high success rate over the task range, outperforming GTF at very long delays (Fig. 3b). When results are aggregated across all delays, EF-BPTT proved not only systematically better when compared to TF-BPTT, but it was also more robust to variations in $\alpha$ (Fig. 3c). This implies that the directionality of the feedback w.r.t. the zero-error manifold is more important for learning in this task than the precise strength of the forcing.

Next, we focused on input-dependent periodic signal generation, another task for which RNN training is known to be nontrivial [48]. Concretely, the network received one analog input (one out of 7 possible values, selected uniformly at random in each trial), constant throughout the trial, and had to generate two sine waves on its two output channels, with a frequency proportional to the input and a fixed phase difference between them (Fig. 3d). The multidimensional output space allows us to test the generality of EF improvements to variations in $N_y$. Moreover, unlike the previous task, feedback is persistent throughout the trial rather than sparse. As before, convergence is defined by MSE; visually accurate reconstruction corresponds to a MSE of order 0.01, but we report the distribution of test MSEs for the full picture (Fig. 3e).

We varied task difficulty by shifting the set of 7 frequencies upward. BPTT MSE increased substantially with increasing mean frequency within the simulated range (Fig. 3e). Surprisingly, teacher forcing destabilized learning, causing very high test MSE values for the entire frequency range.

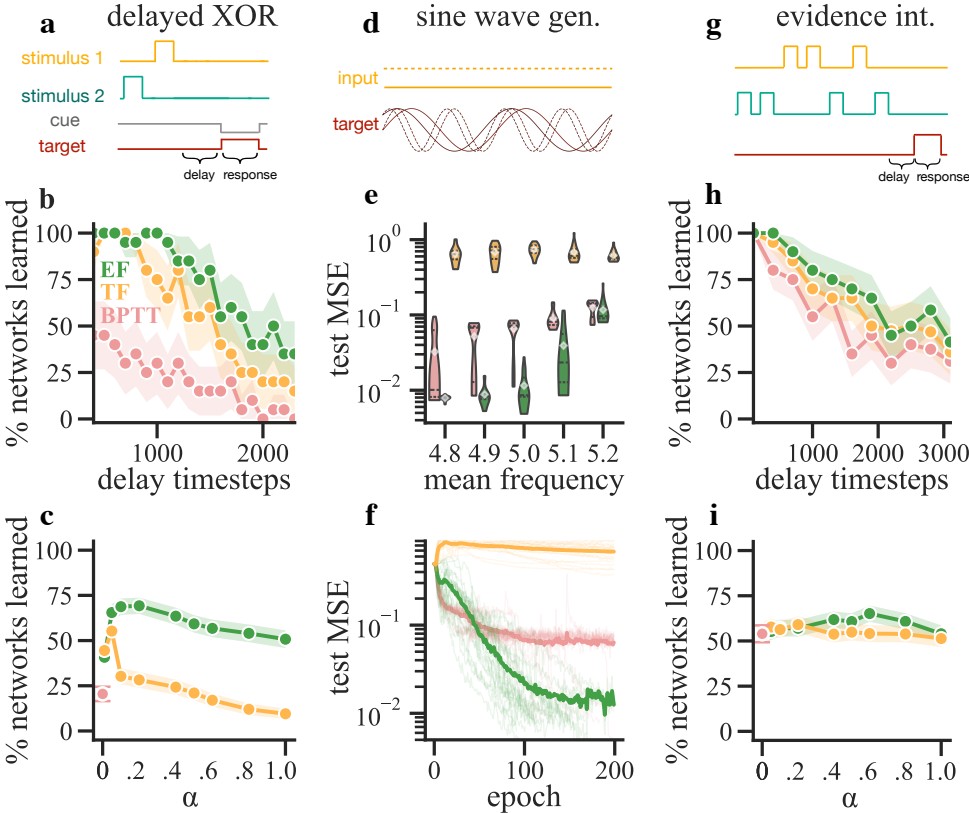

Figure 3: **a)** Delayed XOR schematic. **b)** Proportion of networks that learn the delayed XOR task up to criterion as a function of delay (task difficulty); see main text for details. EF-BPTT and TF-BPTT are shown as EF and TF, respectively, for brevity. **c)** Percentage of networks that learn XOR for different $\alpha$ values, aggregated over every task difficulty. We use $\alpha$ values as small as 0.01. Note that $\alpha = 0$ is equivalent to BPTT for both algorithms. **d)** Schematic of sine generation task: higher inputs (dashed line) require producing sines with higher frequency and the same relative phase. **e)** Mean-squared error violin plots calculated for the sine wave generation task, estimated across 50 different networks for each task setting (x-axis) and for each algorithm (colors as in b). The x-axis denotes the mean frequency of the signals that the network has to generate. **f)** Test MSE during sine generation training for EF-,TF-, and BPTT. Bold lines denote mean over networks. **g)** Schematic of evidence integration task. **h)** As in b) for evidence integration. **i)** As in c) for evidence integration.

Unlike delayed XOR, this task proved more sensitive to a judicious selection of $\alpha$, but error forcing still systematically improved test performance over BPTT.

Lastly, we trained RNNs (40 networks for each setting) on a variant of evidence integration motivated by experiments in rodents [49] (Fig. 3g). In this task, two input channels provide a total of 7 temporally sparse inputs on the 'left' or 'right' side. After an additional delay period (fixed across trials), the network has to output a persistent 1 or -1 during the response period depending on which of the two input channels had more signals. What makes this task an interesting case is the fact that its solution requires a different dynamical systems structure to emerge for efficient learning (line attractor, as opposed to fixed points for XOR, or a set of limit cycles for sine generation) [50, 51]. Similar to the delayed XOR task, forcing is applied only during the response period, with the output of the network unconstrained otherwise. In this case as well EF systematically improves over BPTT, although the differences relative to TF are more modest.

To conclude, across tasks and manipulations, EF-BPTT consistently outperformed TF-BPTT and BPTT, and proved much more robust to the degree of forcing $\alpha$, consistently outperforming BPTT unlike TF-BPTT.

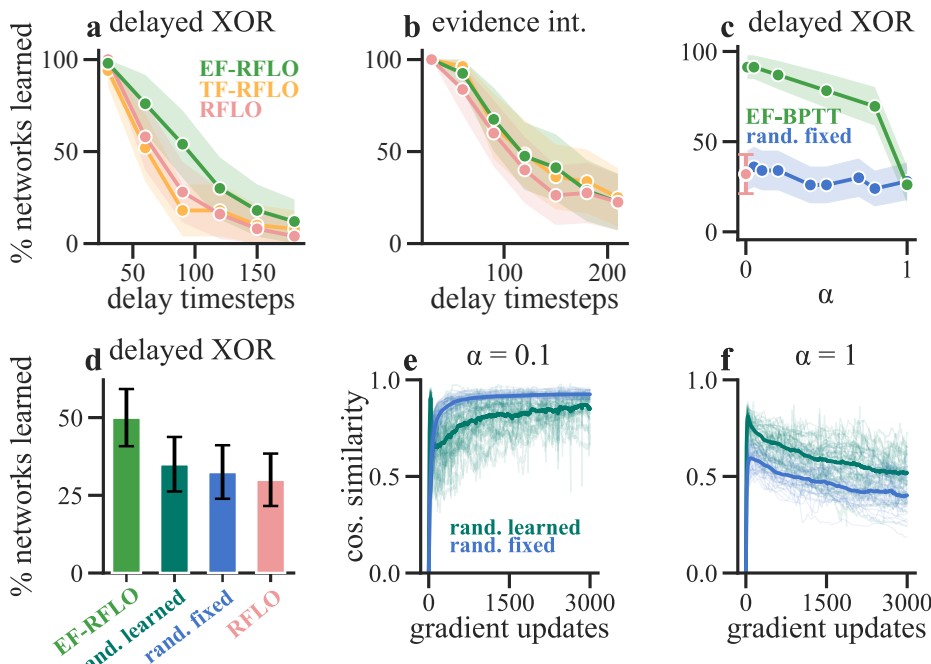

Figure 4: **a)** Percentage of networks learned for a set of networks with increasing delay timesteps (task difficulty), using RFLO. **b)** As in a), but for evidence integration. **c)** Comparison of percentage of networks that learn (-BPTT), for delayed XOR task; with a delay of 1500 time steps. Red denotes BPTT, green denotes EF-BPTT, blue denotes random-fixed projections instead of pseudoinverse. **d)** Comparison of percentage of networks that learn (-RFLO), for delayed XOR task; with a delay of 95 time steps. Red denotes RFLO, green denotes EF-RFLO, blue denotes random-fixed projections, teal green denotes random-learned projections. **e)** Blue: cosine similarity during gradient updates between random-fixed connections and readout weights. Teal-green: between random-learned connections and readout weights. $\alpha = 0.1$. **f)** As in e), but $\alpha = 1.0$.

## 3.2 EF-RFLO

To assess the efficiency of the EF in biologically plausible settings, we equipped RFLO with EF and evaluated learning performance on the delayed XOR and evidence integration tasks, comparing EF-RFLO, TF-RFLO, and RFLO. Despite the demands of temporal credit assignment, RFLO still learned within a limited range of delays (Fig. 4a,b). In the delayed XOR task, adding EF consistently improved performance relative to TF-RFLO and RFLO, though the gains were smaller than with BPTT. All three algorithms performed comparably on the evidence integration task. These results suggest that EF's computational benefits persist under biologically plausible synaptic learning rules.

As explained in Section 2.4, a separate issue is the learning of the feedback forcing weights themselves. For this we consider two different approximations to EF-RFLO that avoid weight transport issues for the error forcing feedback synapses. Instead of using the correct projections (pseudoinverse), we used either random-fixed connections, drawn from the same distribution as the initial readout or random-learned connections, where the learning is achieved by RFLO used on the same loss as for all other network parameters. First, we chose an easy task difficulty for the delayed XOR task, and compared EF-BPTT with random-fixed projections to the idealized version using the pseudoinverse (Fig. 4c). As expected, using random projections severely decreased the performance, supporting the idea that the feedback needs to push the dynamics towards the zero error manifold as opposed to in some arbitrary fixed direction.

When testing the role of adaptive projections for RFLO (Fig. 4d), we found an overall degradation of performance relative to the idealized case but similar or slightly better performance compared to the no feedback (RFLO) scenario.

Investigating the alignment between the readout and error forcing weights during learning reveals an effect similar to feedback alignment in static multilayer perceptron networks [52]. Even with fixed random projections, the cosine similarity between feedback and readout weights steadily increases over time (Fig. 4e,f) because the readout weights become aligned with the fixed random projections. Increasing the $\alpha$ value negatively affects the cosine similarity between the weights; however, alignment is not directly predictive of task performance (Fig. 4 d,e,f). This somewhat paradoxical effect could be explained by the fact that alignment is primarily due to modification of the decoder weights, *not* the feedback weights. Under such conditions, EF suboptimal feedback weights could be effectively constraining the decoder to be in turn suboptimal with respect to the intrinsic network dynamics at initialization, impairing performance.

Overall, we have shown that the error forcing mechanism works in concert with bio-plausible gradient approximation methods (RFLO in particular), although additional work might be needed to fully address the weight transport problem for forcing synapses.

# 4    Discussion

In this study we have introduced *error forcing* (EF), a method capable of reaping the benefits of dynamic error corrections for stabilizing neural dynamics during learning, while still preserving powerful temporal credit assignment capabilities. We have demonstrated empirical improvements of our method over both BPTT and TF-BPTT on several supervised tasks, and were able to justify these improvements in terms of both geometric and Bayesian interpretations of EF. Specifically, error forcing—in contrast to TF—induces a *minimally invasive* intervention on neural dynamics by orthogonally projecting neural activity onto the manifold of optimal outputs. We justified this projection by establishing a connection between EF and variational EM [29], showing that our error corrections can be interpreted as a form of online filtering, where network states are dynamically adjusted based on incoming readout error information. Lastly, while the basic version of EF uses the biologically implausible BPTT algorithm to adjust synaptic weights and suffers from the weight transport problem, we showed that these plausibility issues can be fixed, at the cost of some performance, by a combination of using random or learned error feedback projections in conjunction with the substitution of RFLO for BPTT.

EF can be viewed as a hybridization of Kalman filter-based or control theoretic error correction learning algorithms [25, 26, 53] and TF, theoretically unified by the use of the reparameterization trick. A similar algorithm using a combination of RFLO and driving error feedback has recently been used to model sensorimotor adaptation in response to a virtual reality visuomotor rotation in macaques [27]: our work extends this approach through our theoretical justification for EF, as well as our empirical demonstrations of performance improvements in sparse-feedback supervised learning regimes.

While our analysis has been conducted assuming a linear decoder, EF could be extended to nonlinear (and potentially multilayer) output decoders by using Taylor expansions to locally linearize the output, in analogy to the extended Kalman filter [54] (see Appendix B.4). Similar approaches have been used for feedforward neural networks [21, 24], but it remains an empirical open question whether linear Taylor expansions prove a sufficient approximation for stable use in conjunction with error forcing mechanisms. Such a nonlinear extension could be useful for stabilizing training dynamics in modern state-space models [55, 56], with potential applications to training large-language models [57].

Because of its relationship to Bayesian inference, EF has potential applications for both in-context learning [58] and few-shot learning [59, 60]: namely, the rapid inference of 'errors' allows for EF networks to reach good training performance very rapidly, while test performance is consolidated on a longer timescale through synaptic plasticity. Similarly, though we only used additive error corrections in this study, EF could be combined with hierarchical, compositional Bayesian models to both enable rapid compositional generalization [61, 62] and potentially ameliorate catastrophic forgetting [63].

Lastly, though we have only explored supervised learning in this study, one might expect that most biological motor learning occurs via *reinforcement*. This disconnect could be resolved by using a critic network's expected future reward as a form of short-term supervised objective, as used in active inference [64] and deep Q-learning [65, 66] approaches; in this context, the rapid error correction mechanism provided by error forcing could prove faster and more stable than the Markov-chain

Monte Carlo approach taken by [67] while eliminating the need for the separate parameterized variational posterior distributions used in [68, 69].

To conclude, in this study we have shown that EF is a theoretically justified approach stabilizing temporal credit assignment in sparse supervised feedback regimes; our work opens promising avenues for future research in improving machine learning algorithms, as well as for modeling learning in the brain.

**Broader Impacts:** This study is foundational work devoted to improving optimization of recurrent neural networks, with the goal of furnishing computationally powerful models of learning in the brain. While there are a variety of potentially negative applications of effectively trained recurrent neural network models, extensive modification by third parties would be required to produce such negative impacts on the basis of the insights drawn here.

## Acknowledgments and Disclosure of Funding

We would like to thank members of Savin lab and Neslihan Serap Şengör for insightful discussions and feedback. This work was supported by the National Science Foundation under NSF Award No. 1922658, the Simons Foundation, and a Google faculty award. [CB] is supported in part by the FRQNT Strategic Clusters Program (Centre UNIQUE - Quebec Neuro-AI Research Center).

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

# A  Analyzing the hidden state impact of Error Forcing

In this section, for completeness, we give a step by step derivation explaining how error forcing orthogonally projects the RNN hidden state onto the optimal readout manifold. First, notice that there are infinitely many viable teacher states $\mathbf{r}_t^*$ due to the dimensions of the decoder we consider $\mathbf{W}_\phi$, all of which will produce the optimal readout $\mathbf{y}_t^*$. We can take:

$$\mathbf{r}_t^* = \mathbf{r}_t^{*\min} + \mathbf{z} \tag{14}$$

$$\mathbf{r}_t^{*\min} = \mathbf{W}_\phi^+ \mathbf{y}_t^* \tag{15}$$

$$\mathbf{z} = \alpha_1 \mathbf{v}_1 + \alpha_2 \mathbf{v}_2 + \cdots + \alpha_{N-N_y} \mathbf{v}_{N-N_y} \tag{16}$$

where $\mathbf{v}_i \mid i \in 0, 1, \ldots, N - N_y$ form a basis for the null space of $\mathbf{W}_\phi$, and coefficients $a_i$ are arbitrary scalar coefficients. This solution works for arbitrary coefficients $a_i$ because by definition $\mathbf{W}_\phi \mathbf{z} = 0 \quad \forall a_i$ and $\mathbf{W}_\phi \mathbf{r}_t^{*\min} = \mathbf{y}_t^*$. We will find the teacher state by orthogonally projecting the current state $\mathbf{r}_t$ to the manifold of possible optimal responses. To do so, we can think of the orthogonally projected vector as consisting of two parts: the vector denoting the minimum-norm solution and a second vector denoting the vector needed to get from the minimum-norm solution to the projected solution. The second term can be found by taking the difference vector between the current state and minimum-norm state, and projecting it onto the nullspace of the decoder $\mathbf{W}_\phi$. This can be done by setting the $\mathbf{z}$ in a particular way:

$$\mathbf{P}_{\text{null}} = \mathbf{I} - \mathbf{P}_{\text{row}}, \tag{17}$$

$$\mathbf{P}_{\text{row}} = \mathbf{W}_\phi^+ \mathbf{W}_\phi, \tag{18}$$

$$\mathbf{z} = \mathbf{P}_{\text{null}}(\mathbf{r}_t - \mathbf{r}_t^{*\min}). \tag{19}$$

One can think of this choice of $\mathbf{r}_t^{*\min} + \mathbf{z}$ as the point closest to $\mathbf{r}_t$ that lies on the hyperplane of optimal responses: it is the orthogonal projection of $\mathbf{r}_t$ onto this manifold. Putting it all together:

$$\mathbf{r}_t^* = \mathbf{r}_t^{*\min} + \mathbf{P}_{\text{null}}(\mathbf{r}_t - \mathbf{r}_t^{*\min}) \tag{20}$$

$$= \mathbf{r}_t^{*\min} + [\mathbf{I} - \mathbf{W}_\phi^+ \mathbf{W}_\phi](\mathbf{r}_t - \mathbf{r}_t^{*\min}) \tag{21}$$

Rearranging the terms and using Eq. 15, we have:

$$\mathbf{r}_t^* = [\mathbf{I} - \mathbf{W}_\phi^+ \mathbf{W}_\phi](\mathbf{r}_t - \mathbf{W}_\phi^+ \mathbf{y}_t^*) + \mathbf{W}_\phi^+ \mathbf{y}_t^* \tag{22}$$

$$= [\mathbf{I} - \mathbf{W}_\phi^+ \mathbf{W}_\phi]\mathbf{r}_t + \mathbf{W}_\phi^+ \mathbf{y}_t^* - \mathbf{W}_\phi^+ \mathbf{y}_t^* + \mathbf{W}_\phi^+ \mathbf{W}_\phi \mathbf{W}_\phi^+ \mathbf{y}_t^*. \tag{23}$$

First, notice that,

$$\mathbf{W}_\phi^+ \mathbf{W}_\phi \mathbf{W}_\phi^+ = \mathbf{W}_\phi^+ \tag{24}$$

Therefore,

$$\mathbf{r}_t^* = [\mathbf{I} - \mathbf{W}_\phi^+ \mathbf{W}_\phi]\mathbf{r}_t + \mathbf{W}_\phi^+ \mathbf{y}_t^* - \mathbf{W}_\phi^+ \mathbf{y}_t^* + \mathbf{W}_\phi^+ \mathbf{y}_t^* \tag{25}$$

$$= [\mathbf{I} - \mathbf{W}_\phi^+ \mathbf{W}_\phi]\mathbf{r}_t + \mathbf{W}_\phi^+ \mathbf{y}_t^*. \tag{26}$$

Now we plug this teacher state to the generalized teacher dynamics (Eq. 5):

$$\tilde{\mathbf{r}}_t := (1 - \alpha)\mathbf{r}_t + \alpha \mathbf{r}_t^* \tag{27}$$

$$\Rightarrow \tilde{\mathbf{r}}_t = \left[\mathbf{I} - \alpha \mathbf{W}_\phi^+ \mathbf{W}_\phi\right]\mathbf{r}_t + \alpha \mathbf{W}_\phi^+ \mathbf{y}_t^*. \tag{28}$$

Lastly, one can arrange terms to recover the EF equations we used in the main text:

$$\tilde{\mathbf{r}}_t = \mathbf{r}_t + \alpha \mathbf{W}_\phi^+ (\mathbf{y}_t^* - \mathbf{W}_\phi \mathbf{r}_t) \tag{29}$$

$$= \mathbf{r}_t + \alpha \mathbf{W}_\phi^+ \mathbf{e}_t. \tag{30}$$

In the main text, we apply a 'stop gradient' to $\mathbf{e}_t$, which we justify in Appendix B. Aside from the 'stop gradient' operation, these equations are equivalent to the EF dynamics, which shows that error forcing pushes network activity orthogonally towards the optimal readout manifold. This equation also makes it clear why EF and TF are dynamically same when $N_y \geq N$, because under this condition

$\mathbf{W}_\phi^+ \mathbf{W}_\phi = \mathbf{I}$. It is also worth noting that this approach is similar to how the Kalman Filter was first derived in the seminal paper of Kalman [70], using orthogonal projections.

Here we have compared EF to TF in terms of their *dynamics*. However, since it has been shown [28] that TF can solve the vanishing/exploding gradient problem by damping Jacobians, it is also useful to investigate the differences between EF and TF in terms of their effects on parameter gradients during optimization. Because of its use of stop gradients, EF does not reduce the magnitude of the network dynamics Jacobian in the same way as TF, implying that its performance improvements are due to its guiding influence on network states. However, it is also worthwhile investigating what happens if we *do* let gradients flow through $\mathbf{e}_t$ (explored empirically in Appendix C). When we let the gradients flow, EF only dampens the dimensions of the current state that are in the rowspace of the decoder, replacing this activity with the target (optimal) network state. In this case, the Jacobians become (see Eq. 28):

$$\mathbf{J}_t = \frac{\partial \mathbf{r}_t}{\partial \mathbf{r}_{t-1}} = \frac{\partial \mathbf{r}_t}{\partial \tilde{\mathbf{r}}_{t-1}} \frac{\partial \tilde{\mathbf{r}}_{t-1}}{\partial \mathbf{r}_{t-1}} = \frac{\partial F_\theta(\tilde{\mathbf{r}}_{t-1})}{\partial \tilde{\mathbf{r}}_{t-1}} \frac{\partial \tilde{\mathbf{r}}_{t-1}}{\partial \mathbf{r}_{t-1}} = \tilde{\mathbf{J}}_t(\mathbf{I} - \alpha \mathbf{W}_\phi^+ \mathbf{W}_\phi), \tag{31}$$

This is in contrast to TF, which dampens the activity state in every dimension (decaying activity by a multiplicative factor of $(1 - \alpha)\mathbf{I}$).

## A.1  Network state updates induced by forcing versus gradient updates

The error forcing mechanism involves an optimization process in terms of both network activity states and network parameters; in fact, the error driving mechanism can be viewed as a form of second order optimization derived from Bayesian principles [71] (Appendix B). However, the question stands whether forcing via errors derived from first-order loss gradients could be equally or more effective than using the approach we have taken in this study. In this section we derive a first-order alternative to error forcing and provide empirical comparisons to our approach.

First, consider the loss function:

$$\mathcal{L} = \sum_{t=1}^T \mathcal{L}_t \tag{32}$$

where $\mathcal{L}_t \in \mathbb{R}$ is the instantaneous loss at each time step $t$. As it is commonly done, we assume that the instantaneous loss does not depend on the past activities, given the current activity. The RTRL factorization makes use of the instantaneous loss:

$$\frac{\partial \mathcal{L}}{\partial \theta_i} = \sum_{t=1}^T \frac{\partial \mathcal{L}(t)}{\partial \theta_i} = \sum_{t=1}^T \frac{\partial \mathcal{L}_t}{\partial \mathbf{r}_t} \frac{\partial \mathbf{r}_t}{\partial \theta_i} \tag{33}$$

Where the term $\frac{\partial \mathbf{r}_t}{\partial \theta_i}$ is computed iteratively, using $\frac{\partial \mathbf{r}_{t-1}}{\partial \theta_i}$ from the previous time step. Much work has been done in the iterative term $\frac{\partial \mathbf{r}_t}{\partial \theta_i}$ and it's approximations. However, here we will focus on the first term, $\frac{\partial \mathcal{L}_t}{\partial \mathbf{r}_t}$, because this term defines how network activity $\mathbf{r}_t$ should change to locally improve the loss. Consider instantaneous MSE loss: $\mathcal{L}_t = \frac{1}{2}\mathbf{e}_t^\top \mathbf{e}_t$, where $\mathbf{e}_t = \mathbf{y}_t - \mathbf{y}_t^*$. Using the chain rule,

$$\frac{\partial \mathcal{L}_t}{\partial \mathbf{r}_t} = \frac{\partial \mathcal{L}_t}{\partial \mathbf{e}_t} \frac{\partial \mathbf{e}_t}{\partial \mathbf{y}_t} \frac{\partial \mathbf{y}_t}{\partial \mathbf{r}_t} \tag{34}$$

$$= (\mathbf{e}_t)^\top \mathbf{I}_{N_y} \mathbf{W}_\phi = \mathbf{e}_t^\top \mathbf{W}_\phi \tag{35}$$

Which means that every $\frac{\partial \mathcal{L}_t}{\partial \theta_i}$ term is in the direction of the vector given by Equation 35. Consequently, activity changes induced by synaptic weight changes are in the direction of this vector. One could imagine that nudging network activity states $\mathbf{r}_t$ in the direction of $\frac{\partial \mathcal{L}_t}{\partial \mathbf{r}_t}^\top = \mathbf{W}_\phi^\top \mathbf{e}_t$ could improve performance during training; however, error forcing updates the dynamics via $\mathbf{W}_\phi^+ \mathbf{e}_t$, which is different from Eq. 35, since $\mathbf{W}_\phi^\top \neq \mathbf{W}_\phi^+$. One caveat of using the transpose rather than the pseudoinverse is that the parameter $\alpha$ can no longer be interpreted as an interpolation between the current state and the target state; only the pseudoinverse guarantees the exact target when $\alpha = 1$. We

empirically compared EF and this first-order error forcing approach by sweeping $\alpha$ over a range of values for both methods. We found that EF outperforms the first-order method (Fig. S1c), although for more complex readouts (e.g., multilayer perceptrons, nonlinear functions), first-order gradients may provide a more stable solution for error forcing. In that case, we would use the first-order forcing using the following equation:

$$\tilde{\mathbf{r}}_t = \mathbf{r}_t - \eta g \left( \frac{\partial \mathcal{L}_t}{\partial \mathbf{r}_t}^\top \right) \tag{36}$$

which is equal to $\tilde{\mathbf{r}}_t = \mathbf{r}_t - \eta g(\mathbf{W}_\phi^\top \mathbf{e}_t)$ when a linear decoder combined with a MSE loss is used.

## B  A Bayesian Interpretation of Error Forcing

### B.1  Model definition

To start, consider a noisy dynamical system, with transition function $F_\theta(\cdot) : \mathbb{R}^N \to \mathbb{R}^N$ parameterized by $\theta$:

$$\mathbf{r}_t\big(\boldsymbol{\epsilon}_{0:t}, \mathbf{x}_{0:t}, \theta\big) = F_\theta\Big(\mathbf{r}_{t-1}\big(\boldsymbol{\epsilon}_{0:t-1}, \mathbf{x}_{0:t-1}, \theta\big), \mathbf{x}_t\Big) + \boldsymbol{\epsilon}_t, \tag{37}$$

where $\boldsymbol{\epsilon}_t \sim \mathcal{N}(0, \sigma_\epsilon^2 \mathbf{I})$. Notice that here we define the network dynamics treating the errors $\boldsymbol{\epsilon}_t$ as random variables, as opposed to treating $\mathbf{r}_t$ as a random variable. This is an example of the *reparameterization trick* [42], which will ultimately allow us to do long-term gradient-based credit assignment, as opposed to the traditional EM algorithm for Markovian state-space models, which typically uses temporally 'local' updates [72], relying on the inference distribution to implicitly perform temporal credit assignment. The network predicts targets $\mathbf{y}_t \in \mathbb{R}^{N_y}$ via:

$$\mathbf{y}_t = \mathbf{W}_\phi \mathbf{r}_t\big(\boldsymbol{\epsilon}_{0:t}, \mathbf{x}_{0:t}, \theta\big) + \boldsymbol{\eta}_t, \tag{38}$$

where $\boldsymbol{\eta}_t \sim \mathcal{N}(0, \sigma_{\boldsymbol{\eta}}^2 \mathbf{I})$ and $\mathbf{W}_\phi$ is a $N_y \times N$ matrix, so that we have:

$$p(\mathbf{y}_t | \boldsymbol{\epsilon}_{0:t}, \mathbf{x}_{0:t}) = \mathcal{N}\Big(\mathbf{W}_\phi \mathbf{r}_t\big(\boldsymbol{\epsilon}_{0:t}, \mathbf{x}_{0:t}, \theta\big), \sigma_{\boldsymbol{\eta}}^2 \mathbf{I}\Big). \tag{39}$$

These equations collectively define a stimulus-conditioned generative model for the output:

$$p(\mathbf{y}_{0:T} | \mathbf{x}_{0:T}; \theta) = \int \Big[ \prod_{t=1}^\top p(\mathbf{y}_t, \boldsymbol{\epsilon}_t | \boldsymbol{\epsilon}_{0:t-1}, \mathbf{x}_{0:t-1}; \theta) \Big] p(\mathbf{y}_0, \boldsymbol{\epsilon}_0 | \mathbf{x}_0; \theta) d\boldsymbol{\epsilon}_{0:T}, \tag{40}$$

where $p(\mathbf{y}_0, \boldsymbol{\epsilon}_0 | \mathbf{x}_0; \theta)$ specifies the network's initial conditions and output predictions, and $\sigma_{\boldsymbol{\eta}}^2$ and $\sigma_\epsilon^2$ are free hyperparameters that will ultimately control—via the ratio $\frac{\sigma_{\boldsymbol{\eta}}^2}{\sigma_\epsilon^2}$—how much network states are corrected by output errors, very similar in function to the Kalman gain [54].

### B.2  A review of variational model fitting

Naively, to fit this generative model to a distribution of outputs, one would perform maximum likelihood estimation by minimizing the Kullback-Leibler divergence between the data distribution and the generative model, taking the loss to be:

$$\mathcal{L}_{MLE}(\theta) = KL\Big[p_d(\mathbf{y}_{0:T} | \mathbf{x}_{0:T}) || p(\mathbf{y}_{0:T} | \mathbf{x}_{0:T}; \theta)\Big]. \tag{41}$$

For latent variable models, this loss is difficult to evaluate analytically, because of the intractable integral in Eq. 40 [42, 73]. To resolve this issue, variational methods minimize an upper bound on $\mathcal{L}_{MLE}(\theta)$, called the variational free energy [29], which has better computational properties:

$$\mathcal{L}_{MLE}(\theta) \leq KL\Big[p_d(\mathbf{y}_{0:T} | \mathbf{x}_{0:T}) || p(\mathbf{y}_{0:T} | \mathbf{x}_{0:T}; \theta)\Big]$$

$$+ \mathbb{E}_{p_d(\mathbf{y}_{0:T} | \mathbf{x}_{0:T})} \Big[ KL\Big[ q(\boldsymbol{\epsilon}_{0:T} | \mathbf{x}_{0:T}, \mathbf{y}_{0:T}) || p(\boldsymbol{\epsilon}_{0:T} | \mathbf{x}_{0:T}, \mathbf{y}_{0:T}; \theta) \Big] \Big] \tag{42}$$

$$= KL\Big[ p_d(\mathbf{y}_{0:T} | \mathbf{x}_{0:T}) q(\boldsymbol{\epsilon}_{0:T} | \mathbf{y}_{0:T}, \mathbf{x}_{0:T}) || p(\mathbf{y}_{0:T}, \boldsymbol{\epsilon}_{0:T} | \mathbf{x}_{0:T}; \theta) \Big] \tag{43}$$

$$= \mathcal{L}_{FE}(\theta), \tag{44}$$

where $q(\boldsymbol{\epsilon}_{0:T}|\mathbf{x}_{0:T}, \mathbf{y}_{0:T})$ is a variational inference distribution over the errors $\boldsymbol{\epsilon}_{0:T}$ that we also optimize, and the inequality is simply due to the fact that the KL divergence is strictly positive. The advantage of this approach is that $\mathcal{L}_{FE}(\theta)$ is a function of the joint distribution $p(\mathbf{y}_{0:T}, \boldsymbol{\epsilon}_{0:T}|\mathbf{x}_{0:T}; \theta)$, which unlike the marginal distribution $p(\mathbf{y}_{0:T}|\mathbf{x}_{0:T}; \theta)$ does not require computing an intractable integral. Practically, one selects an appropriate variational inference distribution $q$, and approximates this loss with Monte Carlo samples from the joint distribution $p_d(\mathbf{y}_{0:T}|\mathbf{x}_{0:T})q(\boldsymbol{\epsilon}_{0:T}|\mathbf{y}_{0:T}, \mathbf{x}_{0:T})$, updating the generative model $p(\mathbf{y}_{0:T}, \boldsymbol{\epsilon}_{0:T}|\mathbf{x}_{0:T}; \theta)$ via maximum likelihood estimation.

To select a distribution $q$, one typically minimizes $\mathcal{L}_{FE}$ with respect to some parametric family of distributions (as is the case for variational autoencoders [42]), or one simply notes that $\mathcal{L}_{FE}$ is minimized when one takes $q = p(\boldsymbol{\epsilon}_{0:T}|\mathbf{x}_{0:T}, \mathbf{y}_{0:T}; \theta_{old})$, where $\theta_{old}$ is treated as a constant with respect to subsequent optimization (as is done in variational Expectation-Maximization [29] and gradient EM [74]). This results in a coordinate descent procedure, where one alternates between optimizing $q$ and $\theta$.

### B.3 Error Forcing as approximate gradient EM

In this section, we will show that Error Forcing can be viewed as an approximation of this latter approach. While it would be convenient to take $q = p(\boldsymbol{\epsilon}_{0:T}|\mathbf{x}_{0:T}, \mathbf{y}_{0:T}; \theta_{old})$, inference for arbitrary dynamical systems typically requires a temporally causal *filtering* procedure, followed by a temporally acausal *smoothing* procedure, as required by the forward-backward algorithm [75]: smoothing prevents online learning, because it requires storing a record of past network states. This is much the same problem that BPTT faces, making it unclear how either algorithm could be implemented by a biological system (in contrast to RFLO, for instance). Furthermore, optimal filtering itself for Gaussian state space models requires tracking large covariance matrices for the inferred $\boldsymbol{\epsilon}_t$ variables; it is again unclear how a biological system could store and perform nonlocal computations (like matrix inversion) on $N^2$ state variables.

As an approximation, we will instead do only filtering of maximum *a posteriori* (MAP) estimates. In variational inference terms, this amounts to taking:

$$q(\boldsymbol{\epsilon}_{0:T}|\mathbf{y}_{0:T}, \mathbf{x}_{0:T}) = \prod_{t=0}^{T} \delta(\boldsymbol{\epsilon}_t^*) \tag{45}$$

$$\boldsymbol{\epsilon}_t^* = \operatorname*{argmax}_{\boldsymbol{\epsilon}_t} p(\boldsymbol{\epsilon}_t|\mathbf{y}_{0:t}, \mathbf{x}_{0:t}, \boldsymbol{\epsilon}_{0:t-1}^*; \theta) \tag{46}$$

where $\delta(\cdot)$ is a Dirac delta distribution. Thus, at each time step, we will infer the appropriate error correction $\boldsymbol{\epsilon}_t^*$, given the previous network state, errors, stimuli, and observations, but we will not adjust our error estimate based on future information. After computing the estimated error correction terms $\boldsymbol{\epsilon}_{0:T}^*$ for a sequence, we then update parameters for the recurrent network via gradient descent on Eq. 52. It is important to stress that this approach is an *approximation*: because our maximization with respect to $\boldsymbol{\epsilon}_t$ is performed in a temporally greedy way, it cannot be viewed as global optimization of $\mathcal{L}_{FE}$ as would be required for proper MAP estimation. However, empirically our approach is still quite effective, and the absence of smoothing allows for exclusively online error correction.

Given the overall framework, the question now becomes: how do we calculate the filtered posterior $p(\boldsymbol{\epsilon}_t|\mathbf{y}_{0:t}, \mathbf{x}_{0:t}, \boldsymbol{\epsilon}_{0:t-1}^*; \theta)$, and maximize it with respect to $\boldsymbol{\epsilon}_t$? Since, within our generative model, conditioned on $\boldsymbol{\epsilon}_{0:t-1}^*$, $\mathbf{x}_{0:t}$, and $\mathbf{y}_{0:t-1}$, the current time step's target output $\mathbf{y}_t$ is a linear function of $\mathbf{e}_t$, the joint distribution $p(\boldsymbol{\epsilon}_t, \mathbf{y}_t|\mathbf{y}_{0:t-1}, \mathbf{x}_{0:t}, \boldsymbol{\epsilon}_{0:t-1}^*; \theta)$ is a multivariate Gaussian, with mean and covariance given by:

$$\boldsymbol{\mu} = \begin{pmatrix} \boldsymbol{\mu}_{\boldsymbol{\epsilon}} \\ \boldsymbol{\mu}_{\mathbf{y}} \end{pmatrix} = \begin{pmatrix} 0 \\ \mathbf{W}_\phi \bar{\mathbf{r}}_t \end{pmatrix}, \ \boldsymbol{\Sigma} = \begin{pmatrix} \sigma_{\boldsymbol{\epsilon}}^2 \mathbf{I} & \sigma_{\boldsymbol{\epsilon}}^2 \mathbf{W}_\phi^\top \\ \sigma_{\boldsymbol{\epsilon}}^2 \mathbf{W}_\phi & \sigma_{\boldsymbol{\epsilon}}^2 \mathbf{W}_\phi \mathbf{W}_\phi^\top + \sigma_{\boldsymbol{\eta}}^2 \mathbf{I} \end{pmatrix}, \tag{47}$$

where $\bar{\mathbf{r}}_t = f\left(\mathbf{r}_{t-1}(\boldsymbol{\epsilon}_{0:t-1}^*, \mathbf{x}_{0:t-1}, \theta), \mathbf{x}_t; \theta\right)$. Now the filtered posterior amounts to conditioning this multivariate distribution on the observed output $\mathbf{y}_t$. For a multivariate Gaussian the conditioned distribution has an analytic formula [76]:

$$p(\boldsymbol{\epsilon}_t|\boldsymbol{\epsilon}_{0:t-1}^*, \mathbf{x}_{0:t}, \mathbf{y}_{0:t-1}; \theta) = \mathcal{N}(\boldsymbol{\mu}_{\boldsymbol{\epsilon}|\mathbf{y}}, \boldsymbol{\Sigma}_{\boldsymbol{\epsilon}|\mathbf{y}}), \tag{48}$$

where $\boldsymbol{\mu}_{\boldsymbol{\epsilon}|\mathbf{y}}$ and $\boldsymbol{\Sigma}_{\boldsymbol{\epsilon}|\mathbf{y}}$ are given by:

$$\boldsymbol{\mu}_{\boldsymbol{\epsilon}|\mathbf{y}} = \mathbf{W}_\phi^\top \left(\mathbf{W}_\phi \mathbf{W}_\phi^\top + \frac{\sigma_\eta^2}{\sigma_{\boldsymbol{\epsilon}}^2}\mathbf{I}\right)^{-1}(\mathbf{y}_t - \mathbf{W}_\phi \bar{\mathbf{r}}_t) \tag{49}$$

$$\boldsymbol{\Sigma}_{\boldsymbol{\epsilon}|\mathbf{y}} = \sigma_{\boldsymbol{\epsilon}}^2\mathbf{I} - \sigma_{\boldsymbol{\epsilon}}^2\mathbf{W}_\phi^\top \left(\mathbf{W}_\phi \mathbf{W}_\phi^\top + \frac{\sigma_\eta^2}{\sigma_{\boldsymbol{\epsilon}}^2}\mathbf{I}\right)^{-1}\mathbf{W}_\phi \tag{50}$$

Since the filtered posterior is a Gaussian distribution, the maximum *a posteriori* estimate is equal to the mean. Thus we use Eq. 49 as our point estimate $\boldsymbol{\epsilon}_t^*$ of the posterior. This has the functional form of a feedback error multiplied by a regularized pseudoinverse of the decoder $\mathbf{W}_\phi$, so that the latent network state is corrected, at each time point, by error feedback delivered from the output. These equations provide all of the information we need to perform our optimization procedure; while the pseudoinverse calculation may appear biologically implausible, previous work has shown that it can easily be estimated by regression using a local 'delta' learning rule [46]. For a sequence of inputs $\mathbf{x}_{0:T}$ and outputs $\mathbf{y}_{0:T}$, we iteratively compute network states using the equation:

$$\tilde{\mathbf{r}}_t\left(\boldsymbol{\epsilon}_{0:t}^*, \mathbf{x}_{0:t}, \theta\right) = f\left(\tilde{\mathbf{r}}_{t-1}\left(\boldsymbol{\epsilon}_{0:t-1}^*, \mathbf{x}_{0:t-1}, \theta\right), \mathbf{x}_t; \theta\right) + \boldsymbol{\epsilon}_t^* \tag{51}$$

After computing errors $\boldsymbol{\epsilon}_{0:T}^*$ and network states $\tilde{\mathbf{r}}_{0:T}$ for a full sequence, we update parameters using backpropagation through time [77] on the loss (Eq. 52), or for additional biological realism, RFLO [45]. To see how Eq. 52 produces a standard mean-squared error loss, we can substitute in our choice of variational distribution $q$ and remove terms that do not depend on the parameters $\theta$:

$$\mathcal{L}_{FE}(\theta) = KL\left[p_d(\mathbf{y}_{0:T}|\mathbf{x}_{0:T})q(\boldsymbol{\epsilon}_{0:T}|\mathbf{y}_{0:T}, \mathbf{x}_{0:T})||p(\mathbf{y}_{0:T}, \boldsymbol{\epsilon}_{0:T}|\mathbf{x}_{0:T}; \theta)\right] \tag{52}$$

$$\equiv \mathbb{E}_{p_d(\mathbf{y}_{0:T}|\mathbf{x}_{0:T})}\left[-\log p(\mathbf{y}_{0:T}, \boldsymbol{\epsilon}_{0:T}^*|\mathbf{x}_{0:T}) + \log q(\boldsymbol{\epsilon}_{0:T}^*|\mathbf{y}_{0:T}, \mathbf{x}_{0:T})\right] \tag{53}$$

$$\equiv \mathbb{E}_{p_d(\mathbf{y}_{0:T}|\mathbf{x}_{0:T})}\left[-\log p(\mathbf{y}_{0:T}, \boldsymbol{\epsilon}_{0:T}^*|\mathbf{x}_{0:T})\right]. \tag{54}$$

Here the first equivalency follows from the fact that the data distribution $p_d$ does not depend on the parameters $\theta$ and thus functions as an additive constant, and the second equivalence follows from the fact that the entropy of $q$, as a product of Dirac delta distributions, has constant (negative infinity) entropy that does not depend on the value of $\boldsymbol{\epsilon}_{0:T}^*$ or any network parameters (if one does not want to worry about negative infinities, one may simply consider $q$ to be a product of Gaussian distributions with arbitrarily small variance, where the same principles apply). This loss is now just a negative log-likelihood over targets $\mathbf{y}_{0:T}$ and inferred errors $\boldsymbol{\epsilon}_{0:T}$. Substituting in the log-likelihood of a Gaussian distribution and discarding additive constants, we have:

$$\mathcal{L}_{FE}(\theta) \equiv \mathbb{E}_{p_d(\mathbf{y}_{0:T}|\mathbf{x}_{0:T})}\left[\sum_{t=0}^{T}\frac{1}{2\sigma_\eta^2}(\mathbf{y}_t - \mathbf{W}_\phi\tilde{\mathbf{r}}_t(\boldsymbol{\epsilon}_{0:t}^*, \mathbf{x}_{0:t}, \theta))^\top(\mathbf{y}_t - \mathbf{W}_\phi\tilde{\mathbf{r}}_t(\boldsymbol{\epsilon}_{0:t}^*, \mathbf{x}_{0:t}, \theta)) + \frac{1}{2\sigma_{\boldsymbol{\epsilon}}^2}(\boldsymbol{\epsilon}_t^*)^\top(\boldsymbol{\epsilon}_t^*)\right] \tag{55}$$

$$\equiv E_{p_d(\mathbf{y}_{0:T}|\mathbf{x}_{0:T})}\left[\sum_{t=0}^{T}\frac{1}{2\sigma_\eta^2}(\mathbf{y}_t - \mathbf{W}_\phi\tilde{\mathbf{r}}_t(\boldsymbol{\epsilon}_{0:t}^*, \mathbf{x}_{0:t}, \theta))^\top(\mathbf{y}_t - \mathbf{W}_\phi\tilde{\mathbf{r}}_t(\boldsymbol{\epsilon}_{0:t}^*, \mathbf{x}_{0:t}, \theta))\right], \tag{56}$$

where we may discard the final term because it has no dependence on $\theta$. Therefore, despite its apparent complexity, for our choice of $q$ the loss $\mathcal{L}_{FE}(\theta)$ is no different from a standard mean-squared error, except that the errors $\boldsymbol{\epsilon}_{0:T}^*$ are used to correct internal network dynamics online.

This approach provides a useful balance: we are able to use inferred errors $\boldsymbol{\epsilon}_t^*$ to drive the network into a dynamical regime that is *close* to optimal performance before updating parameters, which has been shown to provide benefits for learning in chaotic recurrent networks [28, 78] and in deep networks [79]; simultaneously, we preserve temporal dependencies between parameters in our recurrent network, enabling the use of BPTT and its variants, which have been shown to be very powerful for fitting complex time series data distributions [80, 69]. To conclude, we can now see that Error Forcing functions as an optimization algorithm (and generalizes to the unforced test-time condition) because it is approximately optimizing an upper bound on $\mathcal{L}_{MLE}(\theta)$, where the driving error signals $\boldsymbol{\epsilon}_{0:T}^*$ function as inferred error corrections similar to those observed in the Kalman Filter.

## B.4 Error forcing with an extended Kalman Filter

In the preceding section we used greedy MAP estimates for $\epsilon_{0:T}$, however these greedy estimates do not account for uncertainty regarding the proper error forcing term at previous time steps. To resolve this issue, we can instead take:

$$q(\epsilon_{0:T}|\mathbf{y}_{0:T}, \mathbf{x}_{0:T}) = \prod_{t=0}^{T} \delta(\epsilon_t^*) \tag{57}$$

$$\epsilon_t^* = \underset{\epsilon_t}{\operatorname{argmax}} \, p(\epsilon_t|\mathbf{y}_{0:t}, \mathbf{x}_{0:t}; \theta) \tag{58}$$

which is to say we now take for $q$ our *filtered* posterior. This approach differs from the previous greedy MAP approach (Eq. 45) in that we do not condition our probability distribution on $\epsilon_{0:t-1}$: our maximization is instead performed by marginalizing over uncertainty in our previous state estimates. This is still an approximation of variational EM, because we are not conditioning $\epsilon_t$ on targets and stimuli from future time points, but it could theoretically improve performance by taking into account uncertainty over previous error forcing terms $\epsilon_{0:t-1}$.

In order to perform this filtered maximization approach, we need, for each timestep, access to $p(\epsilon_t|\mathbf{y}_{0:t}, \mathbf{x}_{0:t}; \theta)$. If our latent transition dynamics (Eq. 37) were linear, this filtered posterior could be computed analytically via Kalman Filtering. However, because the transition dynamics for our noisy RNN are *nonlinear*, we can only perform an approximate inference procedure using the Extended Kalman Filter [54]. For simplicity and consistency with the literature, we will slightly adapt the Extended Kalman Filter equations by writing in terms of $\epsilon_t$ (as opposed to $\mathbf{r}_t$).

Under the Extended Kalman Filter, the filtered posterior $p(\epsilon_t|\mathbf{y}_{0:t}, \mathbf{x}_{0:t}; \theta) \sim N(\bar{\epsilon}_{t|t}, \Sigma_{t|t})$ is normally distributed, with mean $\epsilon_{t|t}$ and covariance $\Sigma_{t|t}$, where the subscript $t|t$ is used to note that the state estimate at time $t$ is conditioned on inputs and targets up to time $t$, whereas the subscript $t|t-1$ will be used to note that the state estimate at time $t$ is conditioned on inputs and targets up to time $t-1$. Estimates are computed iteratively, based on estimates obtained at the previous time $t-1$. First, we define the predicted state estimates given previous state information by rolling forward the network dynamics by one time step:

$$\bar{\mathbf{r}}_{t|t-1} = F_\theta(\mathbf{r}_{t-1} + \bar{\epsilon}_{t-1|t-1}, \mathbf{x}_t) \tag{59}$$

$$\Sigma_{t|t-1} = \mathbf{J}_t \Sigma_{t-1|t-1} \mathbf{J}_t^\top + \sigma_\epsilon^2 \mathbf{I}, \tag{60}$$

where $\Sigma_{t|t-1}$ gives the covariance distribution over latent states before conditioning on targets $\mathbf{y}_t$, and we use $\mathbf{J}_t$ to denote the Jacobian of the transition function evaluated at $\bar{\mathbf{r}}_{t-1|t-1} = \mathbf{r}_{t-1} + \bar{\epsilon}_{t-1|t-1}$, i.e. we have:

$$\mathbf{J}_t = \left. \frac{\partial F_\theta(\mathbf{r}_t, \mathbf{x}_t)}{\partial \mathbf{r}_t} \right|_{\bar{\mathbf{r}}_{t-1|t-1}}. \tag{61}$$

This Jacobian is used in order to linearize network dynamics around the mean estimate of the previous state, in order to preserve the Gaussianity of the inference distribution and the analytic tractability it provides. In terms of these variables, we can then define $\bar{\epsilon}_{t|t}$ and $\Sigma_{t|t}$ as:

$$\bar{\epsilon}_{t|t} = \mathbf{K}_t \left( \mathbf{y} - \mathbf{W}_\phi \bar{\mathbf{r}}_{t|t-1} \right) \tag{62}$$

$$\Sigma_{t|t} = (\mathbf{I} - \mathbf{K}_t \mathbf{W}_\phi) \Sigma_{t|t-1} \tag{63}$$

$$\mathbf{K}_t = \Sigma_{t|t-1} \mathbf{W}_\phi^\top \left( \mathbf{W}_\phi \Sigma_{t|t-1} \mathbf{W}_\phi^T + \sigma_\eta^2 \mathbf{I} \right)^{-1}, \tag{64}$$

where $\mathbf{K}_t$ is the *Kalman gain*. Since the maximum likelihood estimate of a normal distribution is simply its mean, our filtered MAP estimate becomes: $\epsilon_t^* = \bar{\epsilon}_{t|t}$. Comparing Eqs. 62 and 49, we see that our error estimates in the two cases differ because $\Sigma_{t|t-1}$, through the Kalman gain $\mathbf{K}_t$, increases error corrections along dimensions with high latent uncertainty inherited from previous time steps. In the event that we ignore latent uncertainty inherited from previous timesteps, by taking $\Sigma_{t|t-1} = \sigma_\epsilon^2 \mathbf{I}$, these equations are identical–therefore, the error forcing method employed in this paper can be thought of as a simplified approximation of error forcing with an extended Kalman Filter.

Because this latter method requires tracking errors *and* covariance matrices, it is more memory-intensive, requiring records of $N(N+1)$ state variables, and it is unclear how single neurons

could be expected to store such variables in a biologically plausible way. Therefore, as a softer approximation, one could also imagine taking $\boldsymbol{\Sigma}_{t|t-1} \approx \text{diag}\left(\mathbf{J}_t\boldsymbol{\Sigma}_{t-1|t-1}\mathbf{J}_t^\top + \sigma_{\boldsymbol{\epsilon}}^2\mathbf{I}\right)$, so that only $2N$ variables would be required, with each neuron being required to track its own state and its own error-uncertainty. To our knowledge, there is little evidence that cortical neurons maintain or use such an error-uncertainty estimate. All the same, these methods may provide empirical performance improvements relative to Error Forcing in its basic form.

## C   Additional results

We examined the difference between using or not using the stop-gradient in Eq. 9. The principal theoretical difference between using or not using stop gradients is whether the Jacobians are damped in the rowspace of the readout weights (see Eq. 31). Notice this doesn't mean that the Jacobians are always damped in one particular direction during training, because the readout weights are trained through learning and hence they change over learning. We compared the performance of these two approaches on the delayed XOR task (Fig. S1c), where we found that not using stop gradients marginally improved performance. This experiment shows that the benefits of EF mostly come from dynamically guiding the network towards an optimal state, and has less to do with Jacobian damping effects.

Lastly, we trained the networks on the sine wave generation task to examine the effects of choosing different forcing feedback weights: in particular, we compared the pseudoinverse and transpose of the decoder weights (Fig. S1d). We used the sine wave generation task because the differences between using the pseudoinverse and transpose of the readout weights is only evident in the case of multidimensional outputs. Here, we found that using $\mathbf{W}_\phi^+$ produces improvements in performance relative to using the $\mathbf{W}_\phi^\top$, justifying our choice for the feedback weights.

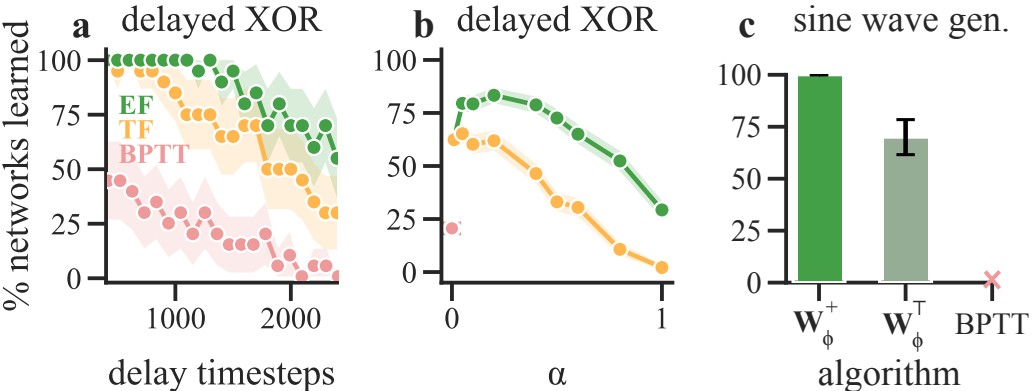

Figure S1: **a)** As in Figure 3b, but without the stop gradient operation. **b)** As in Figure 3c, but without the stop gradient operation. **c)** We trained the sine wave generation task for a specific task difficulty of mean frequency 5.0, using either the pseudoinverse (using $\alpha = 0.01$) of the readout or the transpose.

## D   Details on task and training setup

We used leaky RNNs in the form:

$$\mathbf{r}_t = (1 - \frac{\Delta_t}{\tau})\mathbf{r}_{t-1} + \frac{\Delta_t}{\tau}\mathbf{W}_{\theta_r}\text{tanh}(\mathbf{r}_{t-1}) + \frac{\Delta_t}{\tau}\mathbf{W}_{\theta_x}\mathbf{x}_t + \frac{\Delta_t}{\tau}\mathbf{b} \qquad (65)$$

where $\mathbf{W}_{\theta_r}$ is for recurrent connections, $\mathbf{W}_{\theta_x}$ for input to hidden neurons, $\mathbf{b}$ is the bias, which can be different for each neuron and is trained, and we used a linear decoder as in the main text. For every experiment, we used $\tau = 10$ ms, and $N = 50$ neurons. We never trained the input-to-hidden connections, and always trained the recurrent and output connections with a learning rate 0.001,

using the Adam optimizer, except for the delayed XOR task with -BPTT, where we used learning rate 0.0003. For initialization, we used:

$$\mathbf{W}_{\theta_r}^{ij} \sim \mathcal{N}\left(0, \frac{g^2}{N}\right) \tag{66}$$

where $g$ is the spectral radius, which was set at 1.5. The number of input units $N_x$ depended on the task: it was 3 for delayed XOR (two stimuli, one cue), 2 for evidence integral (two stimuli, no cue), and 1 for sine wave generation (1 stimulus). Inputs were initialized with:

$$\mathbf{W}_{\theta_x}^{ij} \sim \mathcal{N}(0, 1), \tag{67}$$

and finally the decoder weights were initialized with:

$$\mathbf{W}_{\phi}^{ij} \sim \mathcal{N}(0, \frac{1}{N}) \tag{68}$$

As initial values, we used

$$\mathbf{b} \leftarrow \mathbf{0} \tag{69}$$

$$\mathbf{r}_0 \leftarrow \mathbf{0}. \tag{70}$$

We did not treat $\mathbf{r}_0$ as a trainable parameter. We used $L_2$ regularization for neuron activities with a 0.00001 scalar, for evidence integration and the sine wave generation task, which was arbitrarily chosen and was not optimized. For every task, we trained the networks for at most 200 epochs with a batch size of 128. During each epoch, 12 batches were used for testing, i.e., without forcing. We neither clipped the gradients nor used weight decay. Each batch included 512 trials for delayed XOR, 1024 trials for evidence integration, and 2048 trials for sine wave generation. The network was said to have converged if the MSE loss was under 0.1 for delayed XOR and evidence integration, and if it was under 0.01 for sine wave generation, for 10 consecutive epochs. We stopped the training when the network reached convergence according to these criteria.

For delayed XOR, the delay between the stimuli and the stimulus duration was chosen as 10 ms. For evidence integration, it was 5 ms. The delay between the last stimulus and the response period was varied to change the task difficulty; exact values are shown in the main text figures. For the delayed XOR task, the second delay perios was decreased or increased by a maximum of 5 ms, chosen uniformly in a given task difficulty, and the beginning of the response period was indicated with a cue signal. We did not vary the last delay period for the evidence integration task, hence, the task did not require a cue signal. The distribution of stimuli in the evidence integration task for left and right stimuli was chosen uniformly, with a maximum of 7 signals in total. For delayed XOR, input configurations were also chosen uniformly.

For sine wave generation, for a given task difficulty, we randomly (uniformly) picked from 7 discrete frequencies for each trial, and provided the chosen frequency value to the network as a constant input. The networks were required to generate 2 sine waves (output to two different output channels) depending on the given input. These two sine waves were separated by 45 degree phase difference, which was kept fixed for every task difficulty.

The experiments were conducted using an AMD Rome CPU. Each setting was trained using 20 seeds for the delayed XOR and sine wave generation tasks, and 40 seeds for the evidence integration task.

Our code is available at `https://github.com/Savin-Lab-Code/error-forcing`.

