# OpenReview forum: "Error Forcing in Recurrent Neural Networks"
_NeurIPS.cc/2025/Conference — NeurIPS 2025 spotlight_

### Official Review · Reviewer_SdUL · 2025-07-02

**Clarity:** 2
**Significance:** 2
**Originality:** 3
**Rating:** 4
**Confidence:** 4

**Summary:**

To enable learning long-range dependencies in RNNs, this paper proposed Error Forcing (EF) as an algorithm to provide explicit greedy error-based guidance of circuit dynamics along with temporal credit assignment. It builds on top of teacher forcing scheme wherein the updated RNN state is a linear interpolation between with the teacher guided state ($r^*_t$) and the natural RNN state ($r_t$), i.e., the updated RNN state $\tilde{r}_t = (1-\alpha) r_t + \alpha r^*_t$. The RNN non-linear hidden state recursion uses the updated hidden state, i.e., $r_t = F_\theta(r^*_{t-1}, x_t)$. Since the output $y = W_\phi r_t$, for teacher forcing, $r^*_t = W^{-1}_\phi y^*_t$, if the inverse does not exist, one can opt for pseudo inverse $W^{+}_\phi$. However, the issue with teacher forcing is that it constraints the RNN state update and activity heavily towards the teacher guidance.

In this proposed work, Error Forcing uses the target state $r^*_t = r_t + g(W^{+}_\phi  (y^*_t - y_t) )$, this enables exploration of the RNN hidden state along with relaxed guidance from teacher. This work analyzes the proposed error forcing mechanism through the geometric and Bayesian lens. Various empirical experiments have been done to validate the efficacy of the proposed error forcing mechanism on diverse set of tasks. When combined with BPTT, EF-BPTT was able to outperform TF-BPTT and BPTT, for delayed XOR, Sine Wave Generation and evidence integration tasks. Similar observations were achieved when replacing BPTT with RFLO paradigm.

**Questions:**

- In Sec. 3.1, how were the $\alpha$ values chosen for GTF-BPTT and EF-BPTT?
- In Fig 3c, since EF-BPTT reduces to BPTT for $\alpha=0$, why doesn’t the percentage of network learned the same for both these curves at $\alpha=0$ ?
- What is the significance of $\alpha=1$ in the update Eq. (10)? This does not translate to the equivalent teacher forcing update due to the stop gradient operation. Similarly, shouldn’t teacher forcing and error forcing achieve similar performance for $\alpha=1$ regime?
- In evidence int. , BPTT seems to be performing similar to GTF and EF, it’s unclear what advantage these two methods bring over the traditional BPTT on this task.
- In supplementary (C additional results), it seems the experiments without stopping gradients seem to be obtaining decent results. If so, is there any reason to keep the stop-gradient operation in the hidden state update?

**Ethical Concerns:**

["NO or VERY MINOR ethics concerns only"]

**Final Justification:**

Having read author rebuttal and other reviewers' feedback, I believe the proposed method has some merits which allows the RNNs to explore more relaxed trajectory. I agree with some reviewers that the tasks considered in this work for evaluation are toy examples and could be improved to showcase the benefits of the proposed work. I agree that this work has merits and I'll increase my scores to reflect leaning on acceptance on this work.

**Limitations:**

Yes

**Quality:**

2

**Strengths And Weaknesses:**

Strengths:
- Since the teacher guidance is more relaxed compared to the teacher forcing update, Error Forcing (EF) allows the RNN to explore a better hidden state trajectory while still guided with the teacher update.
- Error Forcing outperforms Teacher Forcing in both BPTT and RFLO optimization paradigms in the empirical experiments

Weaknesses:
- The proposed method has only been evaluated with toy experimental setups (such as delayed XOR, sine wave generation, evidence int.). It is unclear how the error forcing would perform on tasks with more complexity and higher dimension input / outputs.
-  In the update equation, Error Forcing proposes to use the stop gradient operation to remove feedback loop back to the teacher guidance. This proposed update seems a bit ad-hoc and has not been explored in detail.

---

> ### Author Rebuttal · Authors · 2025-07-31
>
> We thank the reviewer for the feedback.
>
> **Toy tasks:**  While we acknowledge that the tasks we use are purposefully simple one should note that we deliberately used extremely long delays (more than twice that commonly used in past work, e.g. [1, 2]). Under these harder regimes BPTT succeeds on only ≈25 % of random seeds, while EF still solves >85 % (Figure 1). Since the task is so simple and only time delays increase, it is unequivocally clear that the BPTT failures are directly due to vanishing gradients and so EF owes its performance improvement from mitigating that. To push this point further, we have run further numerical experiments expanding this temporal horizon on delayed XOR task (delay=2500) with BPTT performance %12 +/-4 vs. EF performance %52 +-5, while TF-BPTT is %25 +/-4. This reinforces the idea of EF substantially improving on standard approaches for low dimensional tasks with very long temporal horizons.  While it would be great to expand these results to high dimensional real world applications, we feel that the main goal of this paper is conceptual, with scalability questions best left for a separate publication.
>
> [1] Guillaume Bellec, Franz Scherr, Anand Subramoney, Elias Hajek, Darjan Salaj, Robert Legenstein, and Wolfgang Maass. "A solution to the learning dilemma for recurrent networks of spiking neurons." Nature Communications 11(1): 3625, 2020. doi:10.1038/s41467-020-17236-y
>
> [2] Yuhan Helena Liu, Stephen Smith, Stefan Mihalas, Eric Shea-Brown, and Uygar Sümbül. "Cell-type–specific neuromodulation guides synaptic credit assignment in a spiking neural network." Proceedings of the National Academy of Sciences 118(51): e2111821118, 2021. doi:10.1073/pnas.2111821118
>
> **Stop gradients:** It is important to note that the stop gradient solution is not an ad-hoc choice but rather the formal solution implied by the Bayesian perspective of EF: Because error corrections are dynamically inferred via a coordinate descent procedure on the variational free energy, they are effectively constants with respect to network parameters (hence the use of stop gradients). We do show empirically that EF works with or without the stop gradient. Nonetheless, dropping it comes at the cost of non-local gradient calculations, which makes including the stop gradient still the preferable option from a biological modeling perspective.
>
> **Hyperparameter $\boldsymbol{\alpha}$ selection:** Please see lines 198 and 199 of the text. “The optimal degree of forcing $\alpha$ was determined by a grid search (note that \alpha = 0 reduces EF-BPTT to vanilla BPTT).” This holds for both TF-BPTT and EF-BPTT and for every different task and task difficulty.
>
> **Mismatch in Fig 3c:**  We plot EF‑BPTT only for  $\alpha$ ≥ 0.01; at $\alpha$ = 0 the update is mathematically identical to plain BPTT, so the two curves coincide and we skipped the duplicate point. Even a tiny $\alpha$ (0.01) already gives a clear boost over vanilla BPTT. Note that small $\alpha$ values being already very effective is also noted in the appendix of GTF paper. We will make this point visually clearer in the revised version of Figure 3 and its caption to ensure there is no ambiguity.
>
> **$\boldsymbol{\alpha}$ = 1 limit:** In TF, if pseudoinverse is used, $\alpha$ = 1 gives
>
> $r^{\sim}\_{t}= W^{+}\_{\phi} y^{*}\_{t}$ while for EF, $\alpha = 1$ gives (even without the stop-gradient operation):
>
> $r^{\sim}\_{t} = r_{t} + W^{+}\_{\phi} e_t = r_{t} + W^{+}\_{\phi} (y^{*}\_{t} - y_{t})$ (see equation 9 in the main text)
>
> So these two algorithms are different even when we set $\alpha = 1$, and even when we don’t use the stop-gradient operation.
>
> **Benefits over BPTT in evidence integration task:** We have conducted additional experiments where we trained the BPTT, TF, and EF on evidence integration tasks with limited resources (less trials per epoch). In this regime we saw the importance of the EF algorithm over the other two more prominently: delay timesteps: 1000, EF-BPTT: %84 +/-8, TF-BPTT: %73 +/-8, BPTT: %52 +/-4
>
> We believe that for this task if enough resources are provided the learning algorithms perform similarly, while in the less resource regime EF performs better.

---

### Official Review · Reviewer_XnST · 2025-07-03

**Clarity:** 3
**Significance:** 3
**Originality:** 3
**Rating:** 5
**Confidence:** 3

**Summary:**

The authors introduce Error Forcing, a departure from Teacher Forcing (TF) that focuses on minimal intervention in including error-based learning, while also adding gradient stopping to deal with long-term temporal dependencies.  The authors provide geometric and Bayesian perspectives to explain the methodology, as well as consider more biologically realistic alternatives. They demonstrate the learning model on a variety of tasks, constrasting it against classical BPTT (a subcase of their method) and TF.

**Questions:**

- Could you explain further the distinction between training and testing alluded to at the end of Section 2.3?
- Is there a mismatch at $\alpha = 0$ in Fig 3c with EF-BPTT and BPTT not coinciding?
- What is going on in early training in Fig. 3F, with EF-BPTT being a bit worst at first than BPTT? What explains this difference in behavior?

**Ethical Concerns:**

["NO or VERY MINOR ethics concerns only"]

**Final Justification:**

The authors have addressed my concerns, and I believe the paper is in good enough standing for acceptance. I maintain however my confidence level.

**Limitations:**

yes.

**Paper Formatting Concerns:**

None.

**Quality:**

3

**Strengths And Weaknesses:**

Strengths:
- The motivation for Error forcing as minimal intervention is well-framed.
- The geometric and Bayesian perspectives are complementary and both useful to provide intuition.
- The authors demonstrate their inference methodology across a variety of tasks with varying difficulties.

Weaknesses:
- The argument that less forcing implies more exploration could be made stronger. Theoretical derivations, providing geometric intuition beyond full forcing or incorporating numerical experiments to corroborate would all strengthen the claim.
- The work is fairly skim on the inclusion of other relevant works, especially on online learning from feedback errors.
- The Bayesian perspective is insightful but would benefit from being more carefully detailed. I gather the EF forward pass emulates Kalman filtering (or potentially recursive least squares for static targets) and the variational EM comes from the alternation with the backward pass. These connections feel high-level, however, when in fact they are quite exact. I believe fleshing out these ideas even further and bringing in details from the Appendix could help the reader and strengthen the argument.

---

> ### Author Rebuttal · Authors · 2025-07-31
>
> We thank the reviewer for appreciating that the provided "perspectives are complementary and both useful to provide intuition” and that we evaluated Error Forcing “across a variety of tasks with varying difficulties”. We addressed the raised issues below:
>
> **On the ”more exploration during learning” advantage of EF vs. TF:** We agree that the demonstration of the effect could be more precise; to address this, we performed new numerical tests to empirically estimate the dimensionality of neural activity over learning, directly comparing EF and TF. The new experimental results will be added in the revised version.
>
> **On the inclusion of other relevant works:** We would be happy to discuss the broader online learning from feedback work, however, we are uncertain which specific works the reviewer has in mind. Could you please clarify the works you are referring to?
>
> **More on Bayesian perspective:** The connections to Kalman and EM are indeed formally tight and detailed in the Appendix; we can revise the text to make that even clearer in the main text. We are also adding more information on how to generalize the approach for a tighter treatment of uncertainty and the EF implications of that at the prompting of reviewer j5Yw.
>
> **Train/test:** During training, we apply error forcing to adjust network dynamics, whereas during testing, we do not employ error forcing at all. Naively, one might assume that a network trained in the presence of error forcing might come to ‘rely’ on that error signal, and would not generalize well to unforced conditions. This is also a concern for teacher forcing, which has led to approaches that gradually reduce the amount of forcing required over training [1]. We did not find such annealing empirically necessary to reach good performance for our networks. More importantly, our results formally justify the generalization performance of our trained networks: according to the principles of the variational EM algorithm, Error Forcing maximizes a lower bound on the true log-likelihood by jointly adapting hidden states (corresponding to the E step) and weights (M step); nudging is therefore part of the exact training objective, not an external oracle: the same generative objective is optimised in training and evaluated in testing. The Bayesian perspective on EF thus provides a formal justification for the train-test difference in dynamics.
>
> [1] Jonas Mikhaeil, Zahra Monfared, and Daniel Durstewitz. On the difficulty of learning chaotic dynamics with rnns. In S. Koyejo, S. Mohamed, A. Agarwal, D. Belgrave, K. Cho, and A. Oh, editors, Advances in Neural Information Processing Systems, volume 35, pages 11297–11312. Curran Associates, Inc., 2022.
>
> **Mismatch in Fig 3c:**  We plot EF‑BPTT only for  $\alpha$ ≥ 0.01; at $\alpha$ = 0 the update is mathematically identical to plain BPTT, so the two curves coincide and we skipped the duplicate point. Even a tiny $\alpha$ (0.01) already gives a clear boost over vanilla BPTT. Note that small $\alpha$ values being already very effective is also noted in the appendix of GTF paper. We will make this point visually clearer in the revised version of Figure 3 and its caption to ensure there is no ambiguity.
>
> **Early training in Fig 3f:** Thank you for pointing this out. When investigating the loss curves for other tasks, we did not see the same pattern, so we believe this is an idiosyncratic behavior specific to that numerical experiment.

---

> > ### Comment · Reviewer_XnST · 2025-08-07
> >
> > Thank you to the authors for their response. I am satisfied with the response, and would strongly support including these details in the main text in the revision, as well as the derivations mentioned to reviewer j5Yw alluded to here. I will increase my score, maintaining my confidence.

---

### Official Review · Reviewer_j5Yw · 2025-07-03

**Clarity:** 4
**Significance:** 3
**Originality:** 3
**Rating:** 5
**Confidence:** 4

**Summary:**

This paper proposes a variation on teacher forcing for RNNs, which moves the internal state on each timestep toward a target state on the solution manifold (the set of states yielding zero error). The proposed Error Forcing modifies teacher forcing by picking the target state that minimizes the norm of the adjustment rather than the norm of the target state itself.

**Questions:**

EF and TF both train the model for a different task than it will be tested on. That is, its weights are optimized for a setting where its state is nudged by an oracle on each timestep. How does this affect generalization to the test scenario where there is no nudging? Would the model benefit from being weaned off EF in the latter phases of training (by reducing alpha toward zero)?

The greedy MAP estimation of epsilon makes the Bayesian connection somewhat superficial; it’s just the classic idea that delta-rule learning can be interpreted as the mean update of a conjugate Gaussian prior. If you wanted to more seriously pursue the Bayesian connection you could do a proper (extended) KF and carry along uncertainty in r_t (probably using a diagonal covariance matrix for computation efficiency).

It’s unclear whether the metrics in the experiments are from training or testing — specifically the % networks learned in figs 3bchi and 4abd and MSE in 3e (whereas the caption for 3f says test MSE).

A similar issue comes up in Transformers, specifically causal language modeling where teacher forcing entails that LLMs are typically trained only on one step ahead prediction (e.g., arxiv:2403.06963). This goes beyond the present paper’s scope but I wonder whether EF would be helpful there.

**Ethical Concerns:**

["NO or VERY MINOR ethics concerns only"]

**Final Justification:**

Theoretically strong and conceptually well motivated method. Experiment results are convincing for toy domains.

**Limitations:**

yes

**Quality:**

3

**Strengths And Weaknesses:**

Strengths

- Well motivated variation on the existing teacher forcing; retrospectively this is clearly the right way to do TF
- Principled Bayesian motivation
- Clear experimental advantage over TF and baselines

Weaknesses

- Experiments are in toy domains. Scaling up is especially important for this project because of the nature of the problem it aims to solve (vanishing gradients over long sequences).

---

> ### Author Rebuttal · Authors · 2025-07-31
>
> We thank the reviewer for highlighting both its merits (“clearly the right way to do TF”, “principled Bayesian motivation”) and the areas that would benefit from additional clarification or empirical evidence. Detailed answers below.
>
> **Toy tasks:**  While we acknowledge that the tasks we use are purposefully simple one should note that we deliberately used extremely long delays (more than twice that commonly used in past work, e.g. [1, 2]). Under these harder regimes BPTT succeeds on only ≈25 % of random seeds, while EF still solves >85 % (Figure 3). Since the task is so simple and only time delays increase, it is unequivocally clear that the BPTT failures are directly due to vanishing gradients and so EF owes its performance improvement from mitigating that. To push this point further, we have run further numerical experiments expanding this temporal horizon on delayed XOR task (delay=2500) with BPTT performance %12 +/-4 vs. EF performance %52 +-5, while TF-BPTT is %25 +/-4. This reinforces the idea of EF substantially improving on standard approaches for low dimensional tasks with very long temporal horizons.  While it would be great to expand these results to high dimensional real world applications, we feel that the main goal of this paper is conceptual, with scalability questions best left for a separate publication.
>
> [1] Guillaume Bellec, Franz Scherr, Anand Subramoney, Elias Hajek, Darjan Salaj, Robert Legenstein, and Wolfgang Maass. "A solution to the learning dilemma for recurrent networks of spiking neurons." Nature Communications 11(1): 3625, 2020. doi:10.1038/s41467-020-17236-y
>
> [2] Yuhan Helena Liu, Stephen Smith, Stefan Mihalas, Eric Shea-Brown, and Uygar Sümbül. "Cell-type–specific neuromodulation guides synaptic credit assignment in a spiking neural network." Proceedings of the National Academy of Sciences 118(51): e2111821118, 2021. doi:10.1073/pnas.2111821118
>
> **Training vs test in TF/EF:** While Teacher Forcing (TF) indeed introduces a train–test mismatch, EF maximizes a lower bound on the true log-likelihood by jointly adapting hidden states (corresponding to the E step) and weights (M step); nudging is therefore part of the exact training objective, not an external oracle: the same generative objective is optimised in training and evaluated in testing. The Bayesian perspective on EF thus provides a formal justification for the train-test difference in dynamics.
>
> **Annealing EF:** We did not try it for two reasons: 1) In the original GTF paper annealing always performed worse or the same as the GTF. 2) Given the Bayesian perspective on EF,  $\alpha$ value should not be a hyperparameter but the ratio between the latent noise and the observation noise (i.e. constant over time). Nonetheless, due to Error Forcing’s approximate (variational EM) nature, reducing alpha over the training might still prove practically helpful. This potentially introduces two different hyperparameters, one for the rate of (exponential) decay, the other for the lower bound of the $\alpha$ (potentially zero). In new numerical experiments we found that there always exists a constant $\alpha$ value that performs better than the best performing annealing process. We therefore believe that if necessary, optimizing the $\alpha$ value might be a better approach than optimizing the annealing process. We will add these additional experiments in the revised version of the paper.
>
> **The use of uncertainty in EF updates:** While the MAP approximation is already useful in formally justifying the form of the EF update and training/test distinction as EM, it is quite correct that discarding uncertainty information potentially comes with performance costs.
>
> As the reviewer has noticed, tracking uncertainty through time could be accomplished via an explicit extended Kalman Filter approach, or a variational approximation of EKF that uses diagonal precision matrices for increased computational efficiency. Formally, this translates into latent dimension-specific forcing $\alpha$ values that change dynamically through time (though practically Kalman gains stabilize relatively quickly so one could remove the temporal dependence). This would produce less forcing occurring along axes with low latent uncertainty and more along axes with higher latent uncertainty, at the cost of having to track extra state variables through time. To provide guidance for future work, we will include in the Appendix the full EKM derivation under our error reparameterization, as well as a variational approximation of the extended Kalman Filter using diagonal precision matrices for increased computational efficiency.
>
> **Metrics in the experiments:** All quantitative results in Figs 3 and 4 are always on a test set. We will add “test” to every y-axis label that lacks it, and state explicitly in the captions.
>
> The reference on TF in Transformers is interesting, we will mention it in discussion.

---

> > ### Comment · Reviewer_j5Yw · 2025-08-08
> >
> > Thanks for the thorough replies. They have addressed all my concerns and further convinced me of the theoretical merits. I believe the paper is a clear Accept now.

---

### Decision · Program_Chairs · 2025-09-17

**Decision:**

Accept (spotlight)

**Comment:**

The paper studies learning (non-linear) RNNs with a teacher forcing variant. The paper studies this setting theoretically and empirically and shows strong performance against other RNN learning algorithms.

Strengths: All reviewers and me unambiguously find the paper well written and the results thorough and convincing. We agree that the presented method is a well-motivated extension of teacher forcing, with obvious advantages in hindsight.

Weaknesses: The paper is verified only on toy-ish problems empirically.

I, and all reviewers, find the paper to present strong evidence for their claims and the paper clear and insightful.
Furthermore, in the discussion period, the authors were able to address all open concerns.